# AutoDrop: Training Deep Learning Models with Automatic Learning Rate Drop

**Jing Wang**[1]  **Yunfei Teng**[1]  **Anna Choromanska**[1]

[1]Department of Electrical and Computer Engineering, New York University

## Abstract

Modern deep learning (DL) architectures are trained using variants of the SGD algorithm and typically rely on the user to manually drop the learning rate when the training curve saturates. In this paper, we develop an algorithm, that we call AutoDrop, that realizes the learning rate drop automatically and stems from the properties of the learning dynamics of DL systems. Specifically, it is motivated by the observation that the angular velocity of the model parameters, i.e., the velocity of the changes of the convergence direction, for a fixed learning rate initially increases rapidly and then progresses towards soft saturation. At saturation, the optimizer slows down thus the angular velocity saturation is a good indicator for dropping the learning rate. After the drop, the angular velocity "resets" and follows the pattern described above, increasing again until saturation. AutoDrop is built on this idea and drops the learning rate whenever the angular velocity saturates. The method is simple to implement, computationally cheap, and by design avoids the short-horizon bias problem. We show that AutoDrop achieves favorable performance compared to many different baseline manual and automatic learning rate schedulers, and matches the SOTA performance on all our experiments. On the theoretical front, we claim two contributions: we formulate the learning rate behavior based on the angular velocity and provide general convergence theory for the learning rate schedulers that decrease the learning rate step-wise, rather than continuously as is commonly analyzed.

## 1 INTRODUCTION

As data sets grow in size and complexity, it becomes more difficult to pull useful features from them using hand-crafted feature extractors. For this reason, the DL frameworks [Goodfellow et al., 2016] are now widely popular. DL frameworks process input data using multi-layer networks and automatically find high-quality representations of complex data useful for a particular learning task. Today DL approaches are generally recognized as superior to all alternatives for image [Krizhevsky et al., 2012, He et al., 2016], speech [Abdel-Hamid et al., 2012], and video [Karpathy et al., 2014] recognition, image segmentation [Chen et al., 2016], and natural language processing [Weston et al., 2014]. Furthermore, DL is the primary AI technology at major tech companies like Facebook, Google, Microsoft, and IBM, as well as numerous startups, utilized for various learning tasks such as content filtering, photo management, topic classification, search/ads ranking, video indexing, and copyright detection.

Setting the values and schedules of the hyperparameters for training DL models is computationally expensive and time-consuming, e.g., a deep model with around ten billion parameters requires roughly 500 GPUs to be trained in around two weeks [Shoeybi et al., 2019]. Among all hyperparameters used when training DL models, the learning rate schedule is one of the most important [Jin et al., 2020]. For most SOTA DL architectures, the learning rate is dropped several times during training at epochs chosen by the user, i.e., the learning rate is dropped at the predefined epochs, typically when the training loss is expected to saturate. As modern architectures grow larger, manual hyperparameter tuning becomes impractical. Efficient techniques enabling automatic and online hyperparameter adjustment yield significant resource, time, and cost savings (today the cost of training a single SOTA DL model reaches up to hundreds of thousands of dollars [Peng, 2019]).

The automatic learning rate schedule is an open and important problem – having a simple and effective scheme would be very useful, conceptually and practically. This paper addresses the challenge of developing an automatic method for adjusting the learning rate that works in an online fashion during network training. Our approach looks at the prob-

lem of automatic learning rate schedulers from a different perspective than the prior works. Existing automatic learning rate schedulers [Donini et al., 2020, Yang et al., 2019, Baydin et al., 2018, Franceschi et al., 2017, Retsinas et al., 2022] are gradient-based meta-optimization methods that treat the learning rate as a trainable parameter. They suffer from the short-horizon problem [Wu et al., 2018b], which arises when the optimizer becomes overly greedy and focuses solely on minimizing the loss at the current state. The basis for our approach is rooted in a novel concept. We ask: what are the good descriptors of the learning dynamics of DL systems that can guide the automatic learning rate drop? We find that the angular velocity of the model parameters, defined in the very end of this section, is an excellent indicator of the dynamics of the convergence of an optimizer and can be easily used to trigger the learning rate drop during network training. Our algorithm, Autodrop, drops the learning rate whenever the angular velocity saturates. The resulting algorithm that we obtain is extremely simple, it can be used on the top of any DL optimizer (SGD [Bottou, 1998], momentum SGD [Polyak, 1964], ADAM [Kingma and Ba, 2015], etc.), and enjoys an elegant theoretical framework. Moreover, since AutoDrop decays the learning rate only if the optimizer starts to oscillate around the minima, it avoids the short-horizon bias problem that stigmatizes other automatic learning rate techniques. We empirically demonstrate that our method matches the training of DL models and leads to comparable or better generalization compared to SOTA techniques.

Finally, we claim two important theoretical contributions. Firstly, we formulate the learning rate behavior using our proposed angular velocity model. Secondly, we develope a general convergence proof technique applicable not only supports AutoDrop (Theorem 5.1), but is also applicable to any learning rate schedulers that decrease the learning rate step-wise. Most proofs for gradient-based methods require the learning rate to decrease continuously. Our theorems instead support discrete learning rate drop.

This paper is organized as follows: Section 2 discusses the related work, Section 3 builds an intuition for understanding our algorithm based on simple examples, Section 4 shows our algorithm, Section 5 captures the theoretical guarantees, Section 6 presents experimental results, and Section 7 concludes the paper.

**Definition 1.1** (Angular velocity). Define the angular velocity of model parameters as:

$$\omega_i = \angle(s_i, s_{i-1}), \quad \text{where} \quad s_i = x_{i+1} - x_i \qquad (1)$$

and $x_i$ is the parameter vector in the end of the $i^{\text{th}}$ iteration. The operator $\angle(\cdot, \cdot)$ calculates the angle between two vectors and is defined as:

$$\angle(s_i, s_{i-1}) = \frac{180°}{\pi} \cdot \arccos\left(\frac{s_i^{\mathsf{T}} s_{i-1}}{||s_i||||s_{i-1}|| + \epsilon}\right), \quad (2)$$

where $\epsilon$ is a small positive number preventing the division by zero[1][2].

## 2 RELATED WORK

In this section, we summarize different types of learning rate schedulers and divide them into four main categories. *Scheduling-based methods* rely on a carefully designed learning rate schedules that are tailored to the non-convex nature of the deep learning optimization. Cyclical learning rate (CLR) [Smith, 2017] use the cyclical learning rate pattern to train DL models and apply a triangular learning rate policy in each cycle (that is, first increase and then decrease the learning rate linearly in the cycle) to potentially allow for a more rapid traversal of saddle point plateaus. [Smith and Topin, 2017] extends CLR to super-convergence policy OneCycle with only one triangular cycle. [Li and Arora, 2019] exponentially decreases the learning rate and achieves better performance than the constant learning rate. [Agarwal et al., 2021] utilize Chebyshev polynomials in constructing the Chebyshev learning rate schedule, aimed at accelerating vanilla gradient descent. They illustrate that addressing instability issues results in a fractal ordering of step sizes. Another approach [Pesme et al., 2020] builds on the top of the Convergence-Diagnostic algorithm [Pflug, 1990, Chee and Toulis, 2018] that examines the running average of successive gradients' inner products to develop a stopping criterion for the optimizer. The authors expand this idea to build an automatic learning rate adjustment mechanism relying on decreasing the learning rate when a negative inner product is detected. In [Jin et al., 2020], a Gaussian process surrogate model is employed to link the learning rate and expected validation loss. The approach iteratively updates a posterior distribution of validation loss and dynamically searches for the optimal learning rate based on this posterior. Careful design of an acquisition function and forecasting model is necessary to ensure accurate prediction of the validation loss posterior.

Another group of techniques *hypergradient-based methods* [Donini et al., 2020, Yang et al., 2019, Baydin et al., 2018, Franceschi et al., 2017] that optimize both the model parameters and the learning rate simultaneously. The authors of these methods typically introduce a hypergradient that is defined as a gradient of the validation error with respect to the learning rate schedule. The learning rate is optimized online via gradient descent. These techniques however are quite sensitive to the choice of the hyperparameters. Recently, [Retsinas et al., 2022] has presented a second-order hypergradient method which removes extra hyperparameters from training. However, as indicated in [Wu et al., 2018b], all hypergradient methods are struggling to reach SOTA performance due to the existence of short-horizon

---

[1] $\epsilon$ is omitted in the theoretical derivations.

[2] An interpretation of angular velocity could be found in the Supplement (Section 8).

bias. The reason behind it is that all these methods naturally choose the step size that only minimizes the short-term loss, and thus the optimizer tends to ignore the flat region of an ill-conditioned loss surface. A comprehensive discussion on this matter is included in [Wu et al., 2018b].

*Hyperparameter optimization methods* aim to automatically find a good set of hyperparameters offline. They either build explicit regression models to describe the dependence of target algorithm performance on hyperparameter settings [Hutter et al., 2011], or optimize hyperparameters by performing random search along with using greedy sequential methods based on the expected improvement criterion [Bergstra et al., 2011], or use bandit-based approach for hyperparameter selection [Li et al., 2018]. These techniques can be combined with Bayesian optimization [Falkner et al., 2018, Zela et al., 2018]. Recently, several parallel methods have also been proposed for hyperparameter tuning [Jaderberg et al., 2017, Li et al., 2019, Parker-Holder et al., 2020, Li et al., 2020] as well. The hyperparameter optimization methods are computationally expensive in practice.

Popular *adaptive learning rate optimizers* adjust the learning rate for each parameter individually based on gradient information from previous iterations. AdaGrad [Duchi et al., 2011] proposes to update each parameter using a different learning rate which is proportional to the inverse of the past accumulated squared gradients of the parameter. Thus, the parameters associated with larger accumulated squared gradients have smaller step sizes. This method is enabling the model to learn infrequently occurring features, as these features might be highly informative and discriminative. The major weakness of AdaGrad is that the learning rates continually decrease during training and eventually become too small for the model to learn. Later on, RMSprop [Tieleman et al., 2012] and Adadelta [Zeiler, 2012] were proposed to resolve the issue of diminishing learning rate in AdaGrad. Instead of directly summarizing the past squared gradients, both methods maintain an exponential average of the squared gradients, which is used to scale the learning rate of each parameter. The exponential average of the squared gradients could be considered as an approximation to the second moment of the gradients. One step further, ADAM [Kingma and Ba, 2015] estimates both the first and second moments of the gradients and uses them together to update the parameters. To summarize, adaptive learning rate optimizers adjust the step size for each parameter independently based on the gradient information from past iterations in order to speed up the convergence compared to vanilla SGD. These methods still require a universal learning rate to adjust the overall step sizes. AutoDrop is designed to update automatically this universal learning rate and thus could be applied on the top of this class of optimizers.

Finaly, some works focus on novel strategies involving gradient computations in order to enhance the performance of optimizers, e.g.:, [Cohen et al., 2020, Zhang et al., 2019b,

Izmailov, P. et al., 2018]. In our method, we do not change the computation of gradients but put forward a novel automatic learning rate scheduler. Our AutoDrop can therefore be applied on the top of some of these techniques.

# 3 MOTIVATING EXAMPLE

We analyze the properties of the angular velocity for a noisy quadratic model. While simple, this model is used as a proxy for analyzing neural network optimization [Schaul et al., 2013, Martens and Grosse, 2015, Zhang et al., 2019b].

**Definition 3.1** (Noisy Quadratic Model). We use the same model as in [Zhang et al., 2019b]. The model is represented by the following loss function

$$L(x) = \frac{1}{2}(x-c)^\intercal A(x-c), \qquad (3)$$

where $c \sim N(x^*, \Sigma)$ and both $A$ and $\Sigma$ are diagonal. Without loss of generality, we assume $x^* = 0$.

The update formula for the gradient descent at the step $t+1$ is given as ($\alpha$ is the learning rate):

$$x_{t+1} = x_t - \alpha \nabla L(x_t) = x_t - \alpha A(x_t - c_t), c_t \sim N(0, \Sigma).$$

We optimize noisy quadratic model with $x \in \mathbb{R}^{200}$ and $A = diag(\frac{1}{10}, \frac{2}{10}, ..., \frac{200}{10})$ using Gradient Descent (GD), where $\alpha = [0.06, 0.03, 0.01, 0, 001]$. Figure 1 reveals the following properties:

(P1) **Angular velocity saturation:** the angular velocity curves[3] have the tendency to saturate as the training proceeds, and furthermore when the angular velocity enters the saturation phase, the optimizer slows down its convergence,

(P2) **Angular velocity saturation levels:** i) if the learning rate is large enough such that the algorithm cannot converge to the optimum, the angular velocity saturates at a level between 90 degrees and 120 degrees; ii) as the learning rate decreases, and the algorithm systematically converges closer to the optimum, the angular velocity saturates at progressively lower levels; iii) smaller learning rate leads to a slower saturation of the angular velocity; iv) when the learning rate is low enough such that the algorithm can converge to the optimum, the angular velocity saturates at 90 degrees.

These empirical properties can be theoretically justified as shown in the next theorem. Note that we discuss the bound for the cosine value of angular velocity in Theorem 3.2 because the mapping between the cosine value and the angle is a bijection and the cosine value is more amenable for the quantitative analysis.

---

[3]For the noisy quadratic model, the angular velocity (given in Definition 1.1) is computed with respect to one iteration, rather than an epoch, as for this model there is no notion of the epoch.

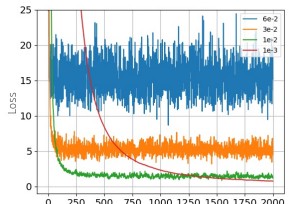 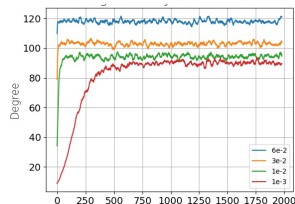 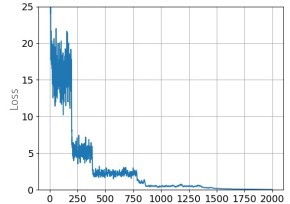 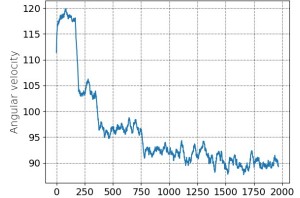

Figure 1: Loss and angular velocity with fixed learning rate for noisy quadratic model.

Figure 2: Loss and angular velocity with dropped learning rate for noisy quadratic model.

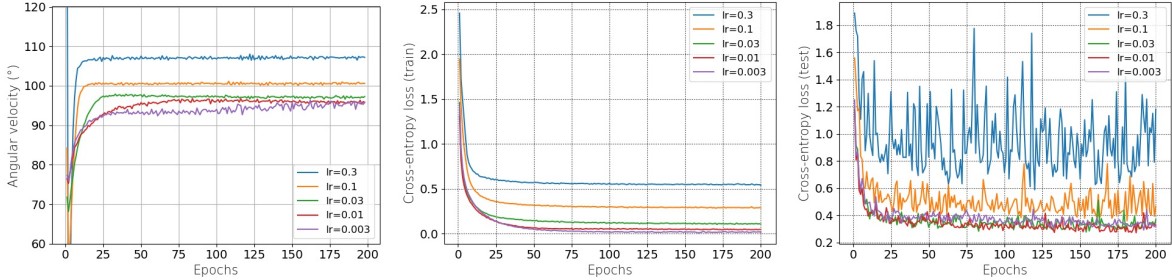

Figure 3: The behavior of the loss and angular velocity for an exemplary DL problem (training ResNet-18 on CIFAR-10). An optimizer is run with different settings of the learning rate $\alpha = [0.3, 0.1, 0.03, 0.01, 0.003]$. Angular velocity is calculated over a single epoch.

**Theorem 3.2.** *Let the $i$-th diagonal terms of matrices $A$ and $\Sigma$ in the noisy quadratic model be given as $a_i$ and $\sigma_i$, respectively. Then, the expected inner product $< s_t, s_{t+1} >$ converges to*

$$I^* = \lim_{t \to \infty} \mathbb{E}[< s_t, s_{t+1} >] = -\alpha^3 \sum_{i=1}^n \frac{a_i^3 \sigma_i^2}{2 - \alpha a_i}. \quad (4)$$

*Moreover, the cosine value of an angle between two consecutive steps $\cos \angle(s_t, s_{t+1})$ satisfies*

$$C^* = \lim_{t \to \infty} \mathbb{E}[cos(\angle(s_t, s_{t+1}))] \approx -\frac{\alpha}{2} \frac{\sum_{i=1}^n \frac{a_i^3 \sigma_i^2}{2-\alpha a_i}}{\sum_{i=1}^n \frac{a_i^2 \sigma_i^2}{2-\alpha a_i}} \quad (5)$$

$$\geq -\frac{\alpha \max_i a_i}{2} \quad (6)$$

$C^* \in [-\frac{1}{2}, 0]$ *and thus* $\lim_{t \to \infty} \angle(s_t, s_{t+1}) \in [90°, 120°]$.

Theorem 3.2 implies that as training proceeds, the angular velocity eventually saturates as stated in property P1. Theorem 3.2 furthermore shows that decreasing the learning rate causes the angle between $s_t$ and $s_{t+1}$ to converge to a smaller value. Also, from Theorem 3.2, $I^* = \lim_{t \to \infty} \mathbb{E}[< s_t, s_{t+1} >] = -\sum_{i=1}^n (\alpha a_i)^3 \sigma_i^2 \left[ \frac{1}{2-\alpha a_i} \right]$. When $\alpha a_i (i = 1, .., n)$ is small enough, $I^*$ can be treated as 0 which implies that $s_t$ is orthogonal to $s_{t+1}$. In other words, the angle between $s_t$ and $s_{t+1}$ converges to 90 degrees for a small enough learning rate. Otherwise, for larger learning rates, this angle saturates above 90 degrees. Furthermore, the limit of cosine angle $C^*$ is approximately larger than $-\frac{1}{2}$, thus the saturation level of angular velocity should be below 120 degrees. This together supports property P2 (in particular this supports points i),ii), and iv); point iii) remains an empirical observation).

We next empirically verified whether these observations carry over to non-convex DL setting on a simple experiment reported in Figure 3. When it comes to the stochastic optimization methods for deep learning methods, the empirical

loss at each iteration is an unbiased estimation of true objective loss with some variance. Therefore, computing the angular velocity defined above with the parameter at each iteration would suffer the problem of variance exploration. To solve this problem, we update Definition 1.1 into Definition 3.3 by adding a sliding window with size $k$ and using the mean of parameters in the window for the computation of the parameter $x_i$.

**Definition 3.3** (Batch angular velocity)**.** Define the angular velocity of model parameters as:

$$\omega_i = \angle(s_i, s_{i-1}), \quad \text{where} \quad s_i = x_{i+1} - x_i \quad (7)$$

and $x_i$ is the mean of the parameter vector in $[ik, (i + 1)k)$ iterations, where $k$ is the size of the sliding window. The operator $\angle(\cdot, \cdot)$ calculates the angle between two vectors and is defined as:

$$\angle(s_i, s_{i-1}) = \frac{180°}{\pi} \cdot \arccos\left(\frac{s_i^\mathsf{T} s_{i-1}}{||s_i|| ||s_{i-1}|| + \epsilon}\right), \quad (8)$$

For analysis simplicity, we take the window size $k$ as the number of iterations in one epoch. Clearly, property P1 holds, whereas property P2 is satisfied partially. In particular conclusion iii) is broken as the angular velocity may not reach 90 degrees.

Property P1 is a key observation underlying our algorithm. The saturation of the angular velocity can potentially guide the drop of the learning rate of the optimization algorithm. In other words, given the lower-bound on the learning rate, each time the angular velocity saturates, the learning algorithm should decrease the learning rate. Tracking the saturation of the angular velocity is more plausible than tracking the saturation of the loss function since, as can be clearly seen in Figure 1, angular velocity curves follow much harder saturation pattern. Also, the loss function does not necessarily need to have a bounded range, as opposed

to the angular velocity. We describe the Algorithm based on property P1 in Section 4. Property P2 is crucial for the theoretical analysis provided in Section 5.

Before moving on to the algorithmic design, we will briefly explain the mechanism that justifies the difference in the behavior between noise quadratic model (NQM) and DL model. The reason DL model does not approach 90 degrees saturation level that instead the NQM can achieve is that the loss surface for NQM is quadratic convex and DL models instead have a highly non-convex loss surfaces, which makes it very difficult to find the global optimum with loss 0. However, note that the saturation levels for the DL model, similarly to NQM, still adhere to the range [90, 120].

Following the above intuition, we implement a simple algorithm for optimizing the noisy quadratic model. The algorithm drops the learning rate by a factor of 2 when the angular velocity saturates (i.e.:, the change of the angular velocity averaged across 20 iterations is smaller than 0.01 degree between 2 consecutive iterations). The initial learning rate was set to 0.06 and the minimal one was set to 0.001. Figure 2 captures the results. It shows that the algorithm that is using the angular velocity to guide the drop of the learning rate indeed converges to the optimum. The aforementioned simple algorithm led us to derive the method for optimizing DL models using automatic learning rate drop that we refer to as AutoDrop. The obtained method is a straightforward extension of the above algorithm and is described in the next section. The extension accommodates the fundamental difference that we observed between noisy quadratic model and the DL model: the fact that in the case of DL models, lower learning rates lead to a larger noise of the angular velocity at saturation.

## 4  ALGORITHM

In this section, we formulate an automatic learning rate schedule algorithm, AutoDrop (Algorithm 1) based on the properties of angular velocity stated in Section 3. The motivation for our method is to drop the learning rate every time the angular velocity saturates. Even though the behavior of angular velocity is much more general compared with the loss (Figure 3), the angular velocity is still fluctuating with variance regarding the choice of different learning rates, which makes setting a hard threshold challenging. We introduce a Gaussian filter to smooth the angular velocity:

$$K(x^*, x_t; \sigma) = \exp\left((x^* - x_t)^2/2\sigma^2\right),$$

where $\sigma$ is the standard deviation of the Gaussian distribution. We define the width of the smoothing buffer as $m$ and denote the buffer as $\mathcal{B}_t = \{x_{t+i}\}_{i=-m/2}^{m/2}$ then the smoothed angular velocity is

$$y_t = Gau(\mathcal{B}; \sigma, m) := \frac{1}{Z(t)} \sum_{i=-m/2}^{m/2} x_t K(x_t, x_{t+i}; \sigma),$$

where $Z(t) = \sum_{i=-m/2}^{m/2} K(x_t, x_{t+i})$. The Gaussian smooth factor at each step $\sigma$ is automatically defined with

---

**Algorithm 1** AutoDrop

**Require:**
  $\alpha_0$ and $\underline{\alpha}$: initial learning rate of the optimizer and its lower bound
  $\rho$: learning rate drop factor
  $x_0$ : initial model parameter vector
  $Gau(\cdot; \sigma, \text{m})$: gaussian filter with smoothing factor $\sigma$, buffer size for smoothing $m$
  $k$ : sliding window size for computing the batch angular velocity.

  $\alpha \leftarrow \alpha_0, s_0 \leftarrow 0, t \leftarrow 0; i \leftarrow 0$
  $\mathcal{B} \leftarrow \{\}$  //Create angular velocity buffer
  **for** t=1,...,T **do**
    Update the parameter $x_t$ with learning rate $\alpha$
    **if** $t \bmod k = 0$ **then**
      $y_i \leftarrow \frac{1}{k} \sum_{t=i*k}^{(i+1)*k-1} x_t$; $\omega_i \leftarrow \angle(y_i, y_{i-1})$;
      $\mathcal{B} = \mathcal{B} \cup \{w_i\}$
      **if** $len(\mathcal{B}) >= 10$ **then**
        $\sigma = min(std(\mathcal{B}), m/2)$
        $\mathcal{C}_i = Gau(\mathcal{B}; \sigma, m)$  //Smooth angular velocity with Gaussian filter
        Drop the first element in buffer $\mathcal{B}$.
      **if** $\mathcal{C}_i - \mathcal{C}_{i-1} < 0.1$ **then**
        $\alpha \leftarrow \max\{\underline{\alpha}, \rho \times \alpha\}$  //Drop $\alpha$
        $\mathcal{B} \leftarrow \{\}$
      $i \leftarrow i + 1$

---

the standard deviation of the current buffer $\mathcal{B}_t$. When the variance of the angular velocity with the current buffer is large, it implies that the angular velocity requires a sharp smooth. Regarding the $\sigma$-rule in statistics (nearly 70% values lie within one standard deviation of the mean), we set an upper bound $\overline{\sigma} = m/2$ for the Gaussian smoothing factor $\sigma$ to avoid too-aggressive smoothing.

The algorithm admits on its input the initial model parameter vector $x_0$, the initial learning rate $\alpha_0$, the value of the smallest permissible learning rate $\underline{\alpha}$, the sliding window size $k$ for computing the batch angular velocity defined in Definition 3.3, the width $m$ of the buffer $\mathcal{B}_t$ used for smoothing of angular velocity and learning rate drop factor $\rho$ ($\rho \in (0, 1)$; each time the learning rate is dropped, it is multiplied by $\rho$). The algorithm triggers the procedure for dropping the learning rate (i.e., multiplied by $\rho$) each time the Gaussian smoothed angular velocity changes by less than the threshold.

The rationale behind dropping the learning rate is not so much to directly accelerate convergence, but rather to help the optimizer that is stuck in the local optimum to escape it. Dropping the learning rate helps DL models escape from current optimum, and finally converge to a better quality one. Note that popular manual learning rate methods (linear learning rate, stepwise learning rate, cosine annealing learning rate, exponential learning rate, etc.) are all decreasing

the learning rate using different mechanisms. Our mechanism is based on the angular velocity. We observed that the saturation of the angular velocity can potentially guide the drop of the learning rate of the optimization algorithm since it is a direct indicator that the optimizer is slowing down, or in other words that the loss function is entering saturation, or in other words that the optimizer is getting stuck in the local optimum. Tracking the saturation of the angular velocity is more plausible than tracking the saturation of the loss function for many reasons (see Figure 1 and 3): i) angular velocity curves follow much harder saturation pattern, ii) the loss function does not necessarily need to have a bounded range, as opposed to the angular velocity, iii) the angular velocity typically enters saturation slightly earlier than the loss function so tracking the angular velocity enables detecting the moment when the optimizer starts to get stuck in local optimum earlier.

Detailed pseudo-code for AutoDrop could be found in Algorithm 1. We further comment on the two fixed conditions $len(\mathcal{B}) > 10$ and $\mathcal{C}_i - \mathcal{C}_{i-1} < 0.1$ in the algorithm. The condition $len(\mathcal{B}) > 10$ means that we will not smooth the angular velocity at the very beginning of the training or right after dropping the learning rate - so this is just a common-sense initial condition since we need to gather a few samples before applying smoothing makes sense. Regarding the condition on $\mathcal{C}_i - \mathcal{C}_{i-1} < 0.1$. Intuitively the threshold for that term should be set to match the standard deviation of the angular velocity. We found that this standard deviation is between 0.1 and 0.25 (see exemplary Table 6 in Supplementary for the ResNet experiment with different learning rates; we observed similar properties for the remaining experiments).

AutoDrop algorithm can be thought of as a meta-scheme that can be put on top of any optimization method for training deep learning models. Thus one can use any optimizer to update model parameters. Next we discuss hyper-parameters used in AutoDrop.

## 4.1 HYPER-PARAMETERS OF AUTODROP

Our method is not hyper-parameter free. note that phrase "automatic" in the paper refers to the techniques that do not need manual adjustments of the learning rate during the optimization process. Other automatic learning rate schedulers that we compare with (TLR and HD) also have hyper-parameters, as well as all manual learning rate techniques. We want to emphasize however that in case of AutoDrop, we keep the hyper-parameters fixed across different experiments, as opposed to for example HD method, and we report ablation studies justifying the settings of the hyper-parameters that we use. Finally, TLR also does not require hyper-parameters to be changed across different experiments, but their performance is inferior to AutoDrop (as will be demonstrated experimentally), and furthermore they perform no ablation studies of their hyper-parameters.

This section discusses the setting of all additional hyper-parameters, over standard optimizers, that AutoDrop introduces: the learning rate drop factor $\rho$, the buffer size $m$ for Gaussian smoothing, and the window size $k$ for computing the batch angular velocity.

Hyperparameters $\rho$ and $m$ are set fixed across all our experiments ($\rho = 0.95$, $m = 10$) and we discuss them first. Note that we also present ablation studies concerning them in the Supplement (Section 10). To ensure that the learning rate does not drop too quickly, $\rho$ should not be too small. Similarly, since excessively large buffer sizes $m$ for Gaussian smoothing leads to over-smoothing and reduced performance, $m$ should not be set to a large value. $\rho = 0.95$, $m = 10$ performed the best in our ablation study on CIFAR10/CIFAR100 tasks. As shown in Section 10, only extreme cases where $\rho$ or $m$ are set to very high values ($\rho = 0.99$, $m = 50$) result in significant changes in the error. In a wide range of settings of these two hyper-parameters we found that the changes of the model performance are not very large, i.e., of the order $2.5\% - 4\%$.

Regarding the sliding window size $k$ used for computing the batch angular velocity, it varies with respect to the size of the training data $N$. Since $k$ decides the frequency of computing the batch angular velocity and we drop the learning rate every time the angular velocity saturates, the learning rate $\alpha_t$ at iteration $t$ for AutoDrop could be simplistically expressed as $\alpha_t = \alpha_0 \rho^{\mathcal{O}(N/k)}$, assuming $\rho$ and $m$ are fixed. Therefore, when the size of the data set $N$ is large, e.g., ImageNet data set has $\sim$1.2M images, the sliding window $k$ should be larger than for smaller data sets, such as CIFAR10 and CIFAR100 tasks that have $\sim$10K data points. We found that $k = 64$ performs well for CIFAR10 and CIFAR100 tasks, while $k = 640$ performs much better for ImageNet. See Table 15 for the ablation study.

| Model | Window size $k$ | | | |
|---|---|---|---|---|
| | $k$=32 | $k$=64 | $k$=128 | $k$=256 |
| ResNet18 CIFAR10 | $5.65_{\pm.15}$ | $\mathbf{4.79_{\pm.99}}$ | $6.08_{\pm.11}$ | $7.41_{\pm.24}$ |
| WRN28x10 CIFAR10 | $4.30_{\pm.13}$ | $\mathbf{3.73_{\pm.07}}$ | $5.77_{\pm.13}$ | $7.36_{\pm.15}$ |
| ResNet34 CIFAR100 | $24.07_{\pm.44}$ | $\mathbf{21.82_{\pm.14}}$ | $23.11_{\pm1.3}$ | $28.33_{\pm.20}$ |
| WRN40x10 CIFAR100 | $20.39_{\pm.08}$ | $\mathbf{19.41_{\pm.10}}$ | $24.49_{\pm.16}$ | $28.79_{\pm.32}$ |
| Model | $k$=64 | $k$=256 | $k$=512 | $k$=640 |
| ResNet18 ImageNet | 39.22 | 31.04 | 29.70 | $\mathbf{29.24}$ |

Table 1: Ablation study for $k$ conducted across different DL models and data sets.

## 5 THEORY

This section theoretically shows that decreasing the learning rate when the angular velocity saturates guarantees the sub-linear convergence rate for SGD and SGD momentum. Moreover, Section 5.1 develops a general convergence proof

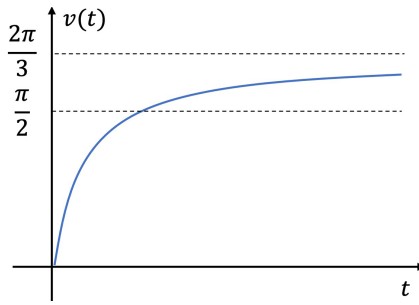

Figure 4: Angular velocity model for a fixed learning rate $\alpha$.

**Algorithm 2** AutoDrop (approximate)

---

**Inputs:** $x_0$: initial weight
**Hyperparameters:** $\{\hat{\alpha}_i\}$: set of learning rates, $v_\alpha(t)$: ang. vel. model, $\tau_0$: init. threshold for the derivative of ang. vel.
Initialize $i = 0, t_0 = 0, t = 0$
**while** $i < n$ **do**
    Update $x_t$ via (9) with learning rate $\alpha_t = \hat{\alpha}_i$.
    **if** $v'_{\hat{\alpha}_i}(t - t_i) \le \tau_i = \min\{\tau_0, \gamma\hat{\alpha}_i/2\}$ **then**
        $i = i + 1; t_i = t$
    $t = t + 1, T = t$
**return** $\{x_t\}_{t=0}^{T-1}$ (T: # iterations)

---

technique that not only supports AutoDrop, but is also applicable to any learning rate schedulers that decrease the learning rate step-wisely.

## 5.1 UNIFIED CONVERGENCE ANALYSIS WITH DISCRETE LEARNING RATE DROP

Firstly, we present a unified theoretical framework that covers the update rule of both SGD and momentum SGD. We refer to these update rules jointly as Unified Momentum (UM) method [Yang et al., 2016]:

$$\text{UM}: \begin{cases} y_{t+1} = x_t - \alpha_t \mathcal{G}(x_t; \xi_t) \\ y_{t+1}^s = x_t - s\alpha_t \mathcal{G}(x_t; \xi_t) \\ x_{t+1} = y_{t+1} + \beta(y_{t+1}^s - y_t^s) \end{cases} \quad (9)$$

where $t$ is the iteration index, $\beta$ is the momentum parameter, $\alpha_t$ is the learning rate at time $t$, $x_t$ is the parameter vector at time $t$, and $\mathcal{G}(x_t; \xi_t)$ is the gradient of the loss function at time $t$ computed for a data mini-batch $\xi_t$. $s$ is the factor that controls the type of optimization method. When $s = 0$ and $s = 1$, UM method is deduced to the heavy-ball and Nestrov (NAG) methods respectively. When $s = 1/(1 - \beta)$, UM method is the vanilla gradient descent method.

Next we prove the convergence of UM methods (Theorem 5.1). The theorem requires some mild constraints on the drop gap $(k_i)$, i.e., number of iterations between two learning rate drops: $i^{\text{th}}$ and $(i+1)^{\text{st}}$. The constraints capture the intuitive argument that extremely lazy changes to the learning rate would bring the scheme close to the constant learning rate method, essentially preventing convergence. Theorem 5.1 accommodates learning settings relying on discrete learning rate drops.

**Theorem 5.1.** *Suppose $f(x)$ is a convex function, $\mathbb{E}\left[\|\mathcal{G}(x; \xi) - \mathbb{E}[\mathcal{G}(x; \xi)]\|\right] \le \delta^2$ and $\|\partial f(x)\| \le G$ for any $x$ and some non-negative $G$. Given a sequence of decreasing learning rates $\{\hat{\alpha}_i\}_{i=-1}^{n-1} \subset (0, 1)$ and a sequence of integers $\{k_i\}_{i=0}^{n-1} \subset \mathbb{N}$ ($n \gg 1$), there exits constants $\kappa_1, \kappa_2$ such that for all $i = 0, ..., n-1$*

$$\hat{\alpha}_i \le (i+2)^{-1}, \ k_i\hat{\alpha}_i \ge \kappa_1, \ k_i\hat{\alpha}_i\hat{\alpha}_{i-1} \le \kappa_2(i+1)^{-1}. \quad (10)$$

*Define a partition $\Pi : 0 = t_0 < t_1 < ... < t_n = T (T = \sum_{i=0}^{n-1} k_i)$. Run UM update defined in Equation 9 for the number of $T$ iterations by setting the learning rate $\alpha_t$ as*

$$\alpha_t = \hat{\alpha}_i, \quad \text{where } t_i \le t < t_{i+1}. \quad (11)$$

*Then the following holds:*

$$\min_{t=0,...,T-1}\{\mathbb{E}[f(x_t) - f(x^*)]\} \le O\left(\log n/\sqrt{n}\right).$$

Note that even in the convex case our analysis is highly non-trivial. All proofs for SGD-based methods require the learning rate to decrease continuously [Wu et al., 2018a, 2019, Gower et al., 2019, Le Roux et al., 2012, Yang et al., 2016, Schmidt et al., 2017, Ramezani-Kebrya et al., 2018, Zhang et al., 2019a] On the other hand, SGD does not converge under a constant learning rate. Discrete learning rate policy (as in AutoDrop) covers the space between constant and continuous learning rate decays. It is non-trivial to see that moving away from a continuous learning rate scheme to a step-wise constant scheme will still sustain the rate of convergence the same as in the continuous learning rate techniques. We also show technical conditions capturing the intuitive argument that extremely lazy changes to the learning rate would bring the step-wise constant learning rate scheme close to the constant learning rate method, essentially preventing convergence. AutoDrop is a discrete learning rate scheduler, which requires new proof techniques compared with the traditional SGD proof scheme. We develop a general proof technique that not only supports AutoDrop, but is also applicable to any learning rate schedulers that decrease the learning rate step-wise. Theorem 5 is therefore universal and of fundamental importance.

In the next section we extend the obtained theorem to our AutoDrop approach.

## 5.2 CONVERGENCE ANALYSIS OF AUTODROP

For a fixed learning rate $\alpha$, we introduce a simplified mathematical model of the behavior of the angular velocity as a function of iterations. The model is defined below (and depicted in Figure 4):

$$v_\alpha(t) = \frac{\pi}{2}(1 + \epsilon\alpha)\left(1 - \frac{1}{\gamma\alpha(t + 1/\gamma\alpha)}\right), \quad (12)$$

where $t$ is the number of iterations, $\epsilon$ and $\gamma$ control the asymptote and curvature of the velocity.

$v_\alpha(t)$ saturates in $\frac{\pi}{2}[1+\epsilon\alpha]$ when $t$ goes to infinity. Note that the given model complies with the property P2 empirically observed and described in Section 3: i) if the learning rate is large enough, the angular velocity saturates at a level larger than $\pi/2$ and smaller than $2\pi/3$; ii) as the learning rate decreases, the angular velocity saturates at progressively lower levels; iii) smaller learning rate leads to a slower saturation of angular velocity; iv) when the learning rate is low enough the angular velocity saturates at $\pi/2$. Let's assume an upper-bound $\alpha_{max}$ for the learning rate. Since the limit of the angular velocity should be between $\pi/2$ and $2\pi/3$, the range of factor $\epsilon$ is set to be $(0, \frac{1}{3\alpha_{max}})$. Finally, Equation (12) is universal and accommodates any saturation level between 90 and 120 degrees, thus the behavior of the DL model from Figure 3 could very well be represented using this Equation.

For the purpose of the theoretical analysis, we drop the learning rate every time the derivative of the angular velocity decreases to a threshold $\tau_i$ (Algorithm 2) instead of detecting whether the change of the angular velocity is small enough (Algorithm 1). Intuitively, when the derivative of the angular velocity is close to zero, we would expect the angular velocity to saturate. The convergence of Algorithm 2 is an approximate version of Algorithm 1. The behavior of the angular velocity and the learning rate for Algorithm 2 is depicted in Figure 5 in Supplementary 11.

**Theorem 5.2.** *Suppose $f(x)$ is a convex function, $\mathbb{E}\left[\|\mathcal{G}(x;\xi) - \mathbb{E}[\mathcal{G}(x;\xi)]\|\right] \leq \delta^2$ and $\|\partial f(x)\| \leq G$ for any $x$ and some non-negative $G$. Given the sequence of the learning rates $\{\hat{\alpha}_i\}_{i=-1}^{n-1}$ such that $\hat{\alpha}_i = (i+1)^{-\frac{2}{3}}$, parameters $\epsilon \in (0, \frac{1}{3\hat{\alpha}_0})$ and $\gamma$ defining the angular velocity model $v_\alpha(t)$ (Equation 12), and the initial threshold $\tau_0$ ($\tau_0 < 2$) for the derivative of the angular velocity, the sequence of weights $\{x_t\}_{t=0}^{T-1}$ generated by Algorithm 2 satisfies*

$$\min_{t=0,\dots,T-1}\{\mathbb{E}[f(x_t) - f(x^*)]\} \leq O\left(\log T/\sqrt{T}\right),$$
*where $\kappa_1 = \frac{\sqrt{\pi}-1}{\gamma}$ and $\kappa_2 = \frac{1}{\gamma}\sqrt{2\pi/3\tau_0}$.*

Theorem 5.2 obtained by extending Theorem 5.1 to the setting accommodating the angular velocity model from Equation 12 guarantees sub-linear convergence rate of Algorithm 2.

# 6 EXPERIMENTAL RESULTS

In this section, we compare the performance of AutoDrop, that automatically adjusts the learning rate, with the SOTA learning rate schedulers for training DL models on the image classification and NLP tasks. In the selection of SOTA baselines, we always choose the best performing strategy reported in the literature for a given data set and architecture. Note that the best performing strategy reported by others relies on manual learning rate drop. For vision tasks the best

performing strategy is referred to as SOTA Baseline and for NLP tasks, this is either ReduceLR or LinearLR in our tables (the references to relevant papers are provided in the text).

We want emphasize that our goal in this paper is to design the automatic learning rate scheduler that could reach the SOTA performance. We do not intend to outperform the SOTA, but rather show that it is possible to design an automatic learning rate scheduler that indeed can match manual schemes that the SOTA relies on. Our method performance-wise matches or outperforms SOTA approach, as will be demonstrated, and wins with all other learning rate schedulers, manual and automatic. So for example existing automatic learning rate schedulers, HD and TLR, lose with SOTA since they suffer from the short-horizon problem, which we by design do not have.

## 6.1 IMAGE CLASSIFICATION

**The CIFAR-10** and **CIFAR-100** data sets[Krizhevsky et al., 2009] consist of 50 K training images, with 10 and 100 different classes respectively. For CIFAR-10 experiments we used a ResNet-18 [He et al., 2016] and a WRN-28x10 [Zagoruyko and Komodakis, 2016] models. For CIFAR-100 experiments we used a ResNet-34 [He et al., 2016] and a WRN-40x10 [Zagoruyko and Komodakis, 2016] models. We do not use the dropout [Srivastava et al., 2014] layers for WRN models in our experiments since this led to better performance. The implementation involving WRN architecture and CIFAR data set relies on publicly available codes[4]. For the above experiments, we refer to [Zhang et al., 2019b] and [Zagoruyko and Komodakis, 2016] for ResNet and WRN models respectively. The ImageNet (ILSVRC-**2012**) data set [Deng et al., 2009] consists of 1.2 M images divided into 1 K categories. We train a ResNet-18 and a ResNet-50[He et al., 2016] model on this data set. We use official model implementation from PyTorch [5].

In our experiments, for the SOTA baseline (the method achieving the best performance on the given data set and model, as reported in the literature) we use the same setting of hyperparameters (including the learning rate schedule) as recommended in the referenced literature. For CLR [Smith, 2017] we test with *triangular2* learning policies by adjusting the *stepsize* (the number of iterations in half a cycle) for different models as recommended by the authors and *OneCycle* policy with only one triangular cycle. For ExpLR [Li and Arora, 2019], we grid search the decay factor from $\gamma = [0.8, 0.9, 0.95, 0.99, 0.999]$. For HD [Baydin et al., 2018] we grid search the hypergradient learning rate $\beta$ from $[10^{-3}, 10^{-4}, 10^{-5}]$ as suggested in the reference paper. For TLR [Retsinas et al., 2022] we set gap $p$ for updating the learning rate as $0.33$ epoch and bound $c = 1/4$, as recommended by the authors. For AutoDrop, we fixed

---

[4]https://github.com/meliketoy/wide-resnet.pytorch
[5]https://pytorch.org/vision/stable/models.html

| Model | HD | TLR | CLR | OneCycle | ExpLR | SOTA Baseline | AutoDrop |
|---|---|---|---|---|---|---|---|
| ResNet18 CIFAR10 | $6.78_{\pm.23}$ | $5.70_{\pm.19}$ | $5.14_{\pm.11}$ | $4.86_{\pm.12}$ | $5.82_{\pm.10}$ | $\mathbf{4.79_{\pm.17}}^{\dagger}$ | $\mathbf{4.79_{\pm.99}}$ |
| WRN28x10 CIFAR10 | $9.12_{\pm.60}$ | $16.70_{\pm2.2}$ | $5.48_{\pm.11}$ | $4.78_{\pm.16}$ | $6.80_{\pm.15}$ | $\mathbf{3.77_{\pm.05}}^{\ddagger}$ | $\mathbf{3.73_{\pm.07}}$ |
| ResNet34 CIFAR100 | $26.89_{\pm1.5}$ | $23.91_{\pm.35}$ | $22.69_{\pm.30}$ | $22.29_{\pm.09}$ | $24.29_{\pm.47}$ | $\mathbf{21.92_{\pm.34}}^{\dagger}$ | $\mathbf{21.82_{\pm.14}}$ |
| WRN40x10 CIFAR100 | $29.32_{\pm.46}$ | $39.54_{\pm.48}$ | $23.61_{\pm.38}$ | $22.60_{\pm.66}$ | $23.32_{\pm.24}$ | $\mathbf{18.96_{\pm0.05}}^{\ddagger}$ | $19.41_{\pm.10}$ |
| ResNet18 ImageNet | 30.43 | 29.81 | 30.48 | 30.67 | 30.10 | $\mathbf{29.74}^{*}$ | $\mathbf{29.246}$ |
| ResNet50 ImageNet | 25.35 | 26.51 | 24.15 | 27.84 | 24.57 | $\mathbf{23.76}^{*}$ | $\mathbf{23.92}$ |

Table 2: Test errors of AutoDrop, SOTA baselines reported in the literature, and baseline manual (CLR, OneCycle, ExpLR) and automatic (HD and TLR) learning rate adjustment algorithms. We run each experiment four times with different random seeds and report the mean and standard deviation of the minimal test error (at the $200^{th}$ epoch for CIFAR10/CIFAR100 and $100^{th}$ epoch for ImageNet). $^{\dagger}$ $^{\ddagger}$ and $^{*}$ follows[Zhang et al., 2019b], [Zagoruyko and Komodakis, 2016] and [He et al., 2016] respectively.

$\rho = 0.95$ and $m = 10$, and searched $k$ for the best one as described in Section 4.1.

Table 2 shows the final test error performance obtained on CIFAR-10, CIFAR-100 and ImageNet datasets and the behavior of the train and test errors/losses and learning rate with epochs for all our experiments is deferred to the Supplement, Section 13.1. Our method shows comparable performance in terms of the test error compared to the manually-tuned SOTA Baseline approaches while automatically selecting the iterations for dropping the learning rate. Simultaneously, AutoDrop was shown superior to manual (CLR, OneCycle, and ExpLR) and automatic (HD and TLR) learning rate adjustment algorithms that were all unable to match the performance of the SOTA baseline.

In Table 3, we also show the the computational time for a single iteration of HD, TLR, SOTA Baseline, and AutoDrop run on the same machine (NVIDIA GeForce GTX 1080 Ti) for different models on different data sets. We use the same batch size of 64 for all methods to have a fair comparison. As you can see the training time per-iteration is practically the same for all methods. Therefore our method does not introduce any additional significant extra computations compared to the existing optimization methods.

| Model\Opt | HD | TLR | SOTA Baseline | AutoDrop |
|---|---|---|---|---|
| WRN28x10 CIFAR10 | 0.21s | 0.23s | 0.20s | 0.20s |
| WRN40x10 CIFAR100 | 0.31s | 0.31s | 0.29s | 0.30s |
| ResNet50 ImageNet | 0.42s | 0.43s | 0.38s | 0.40s |

Table 3: Computational time for a single iteration of HD, TLR, SOTA Baseline, and AutoDrop.

Finally, regarding convergence of the methods, note that the theoretical convergence of our method is shown in the

paper and the rate in theory matches traditional optimizers, such as SGD. The convergence curves are deferred to the Supplement (Section 13). The curves reveal that AutoDrop converges to SOTA performance, unlike other methods. Furthermore, looking at the test error for different methods at different epochs (50, 100, 150, 200) for the exemplary ResNet18/CIFAR10 task (see Table 16 in the Supplement) reveals that AutoDrop reaches comparable performance as SOTA Baseline with sightly faster convergence rate that others cannot attain.

## 6.2 NLP TASKS

**Machine Translation.** A transformer model based on [Vaswani et al., 2017] was trained to translate German to English on the WMT2014 data set [Bojar et al., 2014], using ADAM [Kingma and Ba, 2015] optimizer. The performance of our AutoDrop is compared with ReduceLROnPlateau [Red], HD, and TLR. We train the model for 10K iterations. Table 4 displays the BLEU score obtained on the test data set. The proposed optimizer led to the highest score on the machine translation task. Figure 12 in the Supplementary material 13.2 displays the training curve and shows that AutoDrop also converges faster.

| Model | HD | TLR | ReduceLR | AutoDrop |
|---|---|---|---|---|
| Trans WMT14 | 19.07 | 19.48 | 19.96 | **20.37** |

Table 4: BLUE score of AutoDrop, manual (ReduceLROnPlateau) learning rate and automatic (HD and TLR) learning rate adjustment algorithms on transformer model for WMT2014 data set.

**GLUE Benchmark.** We apply the large language model BERT[Devlin et al., 2018] on the GLUE[Wang et al., 2018] benchmark data set, using ADAM [Kingma and Ba, 2015] optimizer. As is commonly known, the initial increase of

the learning rate during training, which is also known as the "warm-up" phase, plays an important role in the training of large language model. For the GLUE benchmark, we run all methods with and without warm-up and choose the best performer. We compare our Autodrop with the manual learning rate methods: constant learning rate method (ConstLR) and linear learning rate method (LinearLR), and automatic learning rates schemes: TLR and HD. ConstLR is keeping the learning rate constant and LinearLR is reducing it linearly during the training process. And in particular, for AutoDrop and linear and constant learning rate schedulers, adding warm-up improved performance. For the others (HD and TLR), the performance was deteriorated. For constant/linear learning rate, we grid search the learning rate/the peak of learning rate $\alpha$ from $[1e-7, 1e-6, 1e-5]$ and choose the best performer. In Table 5 for each method the best performance is reported. AutoDrop performs much better than automatic learning rate schedulers (HD and TLR) and achieves comparable performance to manual learning rate schedulers (linear and constant learning rate methods).

| GLUE | HD | TLR | LinearLR | ConstLR | AutoDrop |
|------|------|------|------|------|------|
| CoLA | 80.44 | 78.90 | 82.07 | **83.41** | 82.83 |
| MNLI | 78.92 | 81.43 | 83.71 | 83.21 | **83.76** |
| QNLI | 90.46 | 91.17 | 91.54 | 91.32 | **91.74** |
| QQP | 86.42 | 87.33 | **90.51** | 90.48 | 90.04 |
| SST-2 | 91.51 | 91.49 | 92.66 | 91.97 | **92.74** |

Table 5: BLUE score on GLUE benchmark for BERT.

# 7 CONCLUSION

This paper addresses the question: how to relieve the laborious task of tuning the learning rate when training DL models? Our work is motivated by a growing need to develop DL optimization techniques that are more automated in order to increase their scalability and improve the accessibility to DL technology by a wider range of participants. The selection of hyperparameters for training DL models, and especially the learning rate scheduling, is a very hard problem and still remains largely unsolved in the literature. We provide a new algorithm, AutoDrop, for adjusting the learning rate drop during the training of DL models that works online and can be run on top of any DL optimization scheme. AutoDrop has a compelling list of features: it is a simple algorithm to implement and use, it is theoretically well-grounded, it compares favorably to a large cohort of different baseline training approaches, and by design it avoids the short-horizon problem. In our future work, we intend to generalize our approach to automatically schedule other hyper-parameters than the learning rate, such as the momentum term.

### Acknowledgements

The authors acknowledge that the NSF Award #2041872 sponsored the research in this paper. This work was also supported in part by the NYUAD Center for Artificial Intelligence and Robotics, funded by Tamkeen under the NYUAD Research Institute Award CG010.

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

# AutoDrop: Training Deep Learning Models with Automatic Learning Rate Drop
## (Supplementary Material)

## 8 INTERPRETATION OF ANGULAR VELOCITY

The numerator of angular velocity is a dot product of two consecutive gradients. A popular study regarding the meaning of this quantity comes from a hypergradient method Baydin et al. [2018], where the authors discover that the derivative of the loss with respect to the learning rate is closely related to the dot product of the gradients of two consecutive steps. That is:

$$\frac{\partial L(\theta_t)}{\partial \alpha} = -\langle \nabla L(\theta_t), \nabla L(\theta_{t-1}) \rangle,$$

where $L$ is the loss function, $\theta_t$ are the model parameters at step $t$, $\nabla L(\theta_t)$ is the gradient of the loss, and $\alpha$ is the learning rate. From the above formula we can see that the dot product could be used as an indicator for adjusting the learning rate. However, if we adjust the learning too frequently, it will introduce the short-horizon bias problem as we discussed before. Greedily selecting the learning rate only based on the current step may hurt optimizer's performance in the long term. The empirical results of Hypergradient Descent Baydin et al. [2018] (HD) in Table 1 confirm that.

What if we do not allow the learning rate to change until the optimizer saturates? From Theorem 1, we could conclude that for a constant learning rate:

- The expected value of the dot product of the consecutive gradients converges to some value smaller than 0.
- The angular velocities of the gradients converge to some value between 90 and 120 degrees.

This theorem could be interpreted as:

- Under current learning rate, the loss will no longer decrease when the expected value of the dot product of the consecutive gradients/the angular velocity converges.
- The dot product of the consecutive gradients is always smaller than 0 when it converges, which means the learning rate can only decrease, but not increase, after the dot products of the consecutive gradients/angular velocity saturates.

Therefore, the AutoDrop method is designed to detect whether the dot product/angular velocity converges and drop the learning rate then. Furthermore, the angular velocity is easier to track compared to other metrics, like the dot product of consecutive gradients or training loss, because it has much less oscillations when it converges and it is naturally bounded in $[0°, 180°]$. Thus the threshold of saturation for angular velocity becomes easy to determine.

## 9 PROOF FOR THEOREM 3.2

*Proof for Theorem 3.2.* First note that if the learning rate is chosen as specified, then each of the trajectories is a contraction map. By Banach's fixed point theorem, they each have a unique fixed point. Clearly

$$\mathbb{E}^*_{SGD} = \lim_{t \to \infty} \mathbb{E}[x_t] = 0.$$

For the variance we can solve for the fixed points directly. Define $\mathbb{V}^*_{SGD} = \lim_{t \to \infty} \mathbb{V}[x_t]$,

$$\mathbb{V}^*_{SGD} = (I - \gamma A)^2 \mathbb{V}^*_{SGD} + \gamma A^2 \Sigma,$$
$$\Longrightarrow \mathbb{V}^*_{SGD} = \frac{\gamma^2 A^2 \Sigma}{I - (I - \gamma A)^2} = diag(\frac{\alpha^2 a_1^2 \sigma_1^2}{1 - (1 - \alpha a_1)^2}, \cdots, \frac{\alpha^2 a_n^2 \sigma_n^2}{1 - (1 - \alpha a_n)^2}),$$

where $\sigma_i^2$ is the i-th diagonal element of the variance matrix $\Sigma$ of a gaussian noise $c_t$. Because

$$\mathbb{V}_{SGD}^* = \lim_{t \to \infty} \mathbb{V}[x_t] = \lim_{t \to \infty} \mathbb{E}\left[(x_t - \mathbb{E}[x_t])(x_t - \mathbb{E}[x_t])^T\right]$$
$$= \lim_{t \to \infty} \mathbb{E}[x_t x_t^T]$$
$$= diag(\lim_{t \to \infty} \mathbb{E}[x_{t,1}^2], \lim_{t \to \infty} \mathbb{E}[x_{t,2}^2], \cdots, \lim_{t \to \infty} \mathbb{E}[x_{t,n}^2]),$$

we have

$$\lim_{t \to \infty} \mathbb{E}[x_{t,i}^2] = \frac{\alpha^2 a_i^2 \sigma_i^2}{1 - (1 - \alpha a_i)^2} \quad i = 1, \cdots, n. \tag{13}$$

Since $c_t \sim N(0, \Sigma)$,

$$\lim_{t \to \infty} \mathbb{E}[c_{t,i}^2] = \sigma_i^2 \quad i = 1, \cdots, n. \tag{14}$$

The update formula with learning rate $\alpha$ is

$$x_{t+1} = x_t - \alpha \nabla \hat{L}(x_t) = x_t - \alpha A(x_t - c_t), \quad c_t \sim N(0, \Sigma). \tag{15}$$

For the next iteration, the update formula can be written as

$$x_{t+2} = x_{t+1} - \alpha \nabla \hat{L}(x_{t+1}) \tag{16}$$
$$= x_{t+1} - \alpha A(x_{t+1} - c_{t+1}), \quad c_{t+1} \sim N(0, \Sigma)$$
$$= x_{t+1} - \alpha A(x_t - \alpha A(x_t - c_t)), \quad c_t, c_{t+1} \sim N(0, \Sigma)$$
$$= x_{t+1} - \alpha A(x_t - c_{t+1}) + \alpha^2 A^2(x_t - c_t), \quad c_t, c_{t+1} \sim N(0, \Sigma).$$

Define the step at iteration t as $s_t = x_{t+1} - x_t$, then the inner product of two consecutive steps can be written as

$$< s_t, s_{t+1} > = < -\alpha A(x_t - c_t), -\alpha A(x_t - c_{t+1}) + \alpha^2 A^2(x_t - c_t) > \tag{17}$$
$$= \alpha^2 (x_t - c_t)^T A^2 (x_t - c_{t+1}) - \alpha^3 (x_t - c_t)^T A^3 (x_t - c_t)$$
$$= \alpha^2 \left[x_t^T A^2 x_t - x_t^T A^2 c_{t+1} - c_t^T A^2 x_t + c_t^T A^2 c_{t+1} - \alpha x_t^T A^3 x_t + 2\alpha x_t A^3 c_t - \alpha c_t^T A^3 c_t\right].$$

Therefore, the trajectory of the expectation of the inner product converges to

$$I^* = \lim_{t \to \infty} \mathbb{E}[< s_t, s_{t+1} >] = \alpha^2 \left[\lim_{t \to \infty} \mathbb{E}[x_t^T A^2 (I - \alpha A)]x_t - \alpha \lim_{t \to \infty} \mathbb{E}[c_t^T A^3 c_t]\right] \tag{18}$$
$$= \alpha^2 \left[\sum_{i=1}^{n} a_i^2 (1 - \alpha a_i) \lim_{t \to \infty} \mathbb{E}[x_{t,i}^2] - \sum_{i=1}^{n} \alpha a_i^3 \lim_{t \to \infty} \mathbb{E}[c_{t,i}^2]\right]$$
$$= \alpha^2 \sum_{i=1}^{n} \left[a_i^2 (1 - \alpha a_i) \frac{\alpha a_i \sigma_i^2}{2 - \alpha a_i} - \alpha a_i^3 \sigma_i^2\right]$$
$$= \alpha^2 \sum_{i=1}^{n} \alpha a_i^3 \sigma_i^2 \left[\frac{1 - \alpha a_i}{2 - \alpha a_i} - 1\right]$$
$$= -\alpha^3 \sum_{i=1}^{n} \frac{a_i^3 \sigma_i^2}{2 - \alpha a_i}.$$

The norm of step $s_t$ at iteration t is written as

$$\|s_t\|^2 = \|\alpha A(x_t - c_t)\|^2 \tag{19}$$
$$= \alpha^2 (x_t - c_t)^T A^2 (x_t - c_t)$$
$$= \alpha^2 (x_t^T A^2 x_t - 2x_t^T A^2 c_t + c_t^T A^2 c_t).$$

Therefore the trajectory of the expectation of the norm of $s_t$ converges to

$$N^* = \lim_{t\to\infty} \mathbb{E}[\|s_t\|^2] = \alpha^2 \lim_{t\to\infty} \mathbb{E}[x_t^T A^2 x_t] + \alpha^2 \lim_{t\to\infty} \mathbb{E}[c_t^T A^2 c_t] \tag{20}$$

$$= \alpha^2 \sum_{i=1}^n a_i^2 \left( \mathbb{E}[x_{t,i}^2] + \mathbb{E}[c_{t,i}^2] \right)$$

$$= \alpha^2 \sum_{i=1}^n a_i^2 \sigma^2 \left( \frac{\alpha a_i}{2 - \alpha a_i} + 1 \right)$$

$$= 2\alpha^2 \sum_{i=1}^n \frac{a_i^2 \sigma^2}{2 - \alpha a_i}.$$

Here, in order to draw meaningful conclusions we make certain simplifications and proceed by approximating $\mathbb{E}[cos(\angle(s_t, s_{t+1}))] \approx \mathbb{E}[< s_t, s_{t+1} >]/\mathbb{E}[\|s_t\| \|s_{t+1}\|]$.

Because $cos(\angle(s_t, s_{t+1})) = \frac{<s_t, s_{t+1}>}{\|s_t\|\|s_{t+1}\|}$ and $\|s\|_t$ converges when t is large enough, then

$$\lim_{t\to\infty} \mathbb{E}[cos(\angle(s_t, s_{t+1}))] \approx \lim_{t\to\infty} \frac{\mathbb{E}[< s_t, s_{t+1} >]}{\mathbb{E}[\|s_t\|^2]}. \tag{21}$$

Since $I^* = \lim_{t\to\infty} \mathbb{E}[cos(\angle(s_t, s_{t+1}))]$ and $N^* = \lim_{t\to\infty} \mathbb{E}[\|s_t\|^2]$ are both bounded and not equal to 0,

$$\lim_{t\to\infty} \mathbb{E}[cos(\angle(s_t, s_{t+1}))] \approx \frac{\lim_{t\to\infty} \mathbb{E}[< s_t, s_{t+1} >]}{\lim_{t\to\infty} \mathbb{E}[\|s_t\|^2]}. \tag{22}$$

By combining formula (22), (18) and (20), we obtain that the expectation of cosine value converges to

$$C^* = \lim_{t\to\infty} \mathbb{E}[cos(\angle(s_t, s_{t+1}))] \approx \frac{I^*}{N^*} = -\frac{\alpha}{2} \frac{\sum_{i=1}^n \frac{a_i^3 \sigma_i^2}{2 - \alpha a_i}}{\sum_{i=1}^n \frac{a_i^2 \sigma_i^2}{2 - \alpha a_i}} \geq -\frac{\alpha}{2} \max_i a_i \frac{\sum_{i=1}^n \frac{a_i^2 \sigma_i^2}{2 - \alpha a_i}}{\sum_{i=1}^n \frac{a_i^2 \sigma_i^2}{2 - \alpha a_i}} = -\frac{\alpha \max_i a_i}{2} \tag{23}$$

Since $I - \alpha A \succ 0$ implies $\alpha a_i < 1$ for arbitrary $i$, then $C^* \in [-\frac{1}{2}, 0]$ and the angle is between 90 degree to 120 degrees. $\quad\square$

## 10  HYPERPARMATER SETTING FOR AUTODROP

### 10.1  TWO FIXED CONDITIONS IN AUTODROP

We further comment on the two fixed conditions $len(\mathcal{B}) > 10$ and $\mathcal{C}_i - \mathcal{C}_{i-1} < 0.1$ in the algorithm. The condition $len(\mathcal{B}) > 10$ means that we will not smooth the angular velocity at the very beginning of the training or right after dropping the learning rate - so this is just a common-sense initial condition since we need to gather a few samples before applying smoothing makes sense. Regarding the condition on $\mathcal{C}_i - \mathcal{C}_{i-1} < 0.1$. Intuitively the threshold for that term should be set to match the standard deviation of the angular velocity. We found that this standard deviation is between 0.1 and 0.25 (see exemplary Table 6 in Supplementary for the ResNet experiment with different learning rates; we observed similar properties for the remaining experiments).

| ResNet18/CIFAR10 | 1e-1 | 3e-2 | 1e-2 |
|---|---|---|---|
| Standard Deviation | 0.17 | 0.22 | 0.24 |

Table 6: Standard deviation of angular velocity for different learning rate on ResNet18/CIFAR10.

### 10.2  ABLATION STUDY FOR HYPERPARAMETER $\rho$ AND $m$

In this section, we perform ablation study for hyperparameter $\rho$ and $m$ on multiple model settings: ResNet18/CIFAR10, WRN28x10/CIFAR100, ResNet34/CIFAR100 and WRN40x10/CIFAR100. We hyperparamter search $\rho = [0.5, 0.8, 0.9, 0.95, 0.99]$ and $m = [5, 10, 20, 30, 50]$ among different tasks. $\rho = 0.95$, $m = 10$ performs the best among all tasks. In a wide range of hyper-parameter settings that we explore, the changes of the model performance are mild, i.e., of the order $2.5\% - 4\%$.

Therefore, Hyperparameters $\rho$ and $m$ are set fixed across all our experiments ($\rho = 0.95$, $m = 10$).

### 10.2.1 ResNet18-CIFAR10

| Model | Method | $\rho$ | $m$ | $k$ | epoches | Test Error |
|---|---|---|---|---|---|---|
| ResNet18 CIFAR10 | AutoDrop | **0.95** | **10** | **64** | **200** | **4.79 ± 0.99** |
| | | – | 5 | – | 200 | 5.05 ± 0.096 |
| | | – | 20 | – | 200 | 5.50 ± 0.169 |
| | | – | 30 | – | 200 | 6.52 ± 0.092 |
| | | – | 50 | – | 200 | 7.41 ± 0.111 |

Table 7: Ablation study on parameter $m$ for AutoDrop on task ResNet18/CIFAR10.

| Model | Method | $\rho$ | $m$ | $k$ | epoches | Test Error |
|---|---|---|---|---|---|---|
| ResNet18 CIFAR10 | AutoDrop | **0.95** | **10** | **64** | **200** | **4.79 ± 0.99** |
| | | 0.5 | – | – | 200 | 8.85 ± 0.873 |
| | | 0.8 | – | – | 200 | 6.62 ± 0.259 |
| | | 0.9 | – | – | 200 | 5.48 ± 0.040 |
| | | 0.99 | – | – | 200 | 7.65 ± 0.178 |

Table 8: Ablation study on parameter $\rho$ for AutoDrop on task ResNet18/CIFAR10.

### 10.2.2 WRN28x10-CIFAR10

| Model | Method | $\rho$ | $m$ | $k$ | epoches | Test Error |
|---|---|---|---|---|---|---|
| WRN28x10 CIFAR10 | AutoDrop | **0.95** | **10** | **64** | **200** | **3.73 ± 0.07** |
| | | – | 5 | – | 200 | 6.76 ± 0.349 |
| | | – | 20 | – | 200 | 4.63 ± 0.165 |
| | | – | 30 | – | 200 | 4.11 ± 0.137 |
| | | – | 50 | – | 200 | 7.85 ± 0.119 |

Table 9: Ablation study on parameter $m$ for AutoDrop on task WRN28x10/CIFAR10.

| Model | Method | $\rho$ | $m$ | $k$ | epoches | Test Error |
|---|---|---|---|---|---|---|
| WRN28x10 CIFAR10 | AutoDrop | **0.95** | **10** | **64** | **200** | **3.73 ± 0.07** |
| | | 0.5 | – | – | 200 | 6.26 ± 0.283 |
| | | 0.8 | – | – | 200 | 5.07 ± 0.286 |
| | | 0.9 | – | – | 200 | 3.94 ± 0.118 |
| | | 0.99 | – | – | 200 | 7.36 ± 0.021 |

Table 10: Ablation study on parameter $\rho$ for AutoDrop on task WRN28x10/CIFAR10.

### 10.2.3 ResNet34-CIFAR100

| Model | Method | $\rho$ | $m$ | $k$ | epochs | Test Error |
|---|---|---|---|---|---|---|
| ResNet34 CIFAR100 | AutoDrop | **0.95** | **10** | **64** | **200** | **21.82 ± 0.14** |
| | | – | 5 | – | 200 | 22.41 ± 0.187 |
| | | – | 20 | – | 200 | **22.39 ± 0.11** |
| | | – | 30 | – | 200 | 26.09 ± 0.612 |
| | | – | 50 | – | 200 | 28.53± 0.44 |

Table 11: Ablation study on parameter $m$ for AutoDrop on task ResNet34/CIFAR100.

| Model | Method | $\rho$ | $m$ | $k$ | epochs | Test Error |
|---|---|---|---|---|---|---|
| ResNet34 CIFAR100 | AutoDrop | **0.95** | **10** | **64** | **200** | **21.82 ± 0.14** |
| | | 0.5 | – | – | 200 | 30.42 ± 0.430 |
| | | 0.8 | – | – | 200 | 25.71 ± 0.561 |
| | | 0.9 | – | – | 200 | 23.14 ± 0.464 |
| | | 0.99 | – | – | 200 | 30.09 ± 0.192 |

Table 12: Ablation study on parameter $\rho$ for AutoDrop on task ResNet34/CIFAR100.

### 10.2.4 WRN40x10-CIFAR100

| Model | Method | $\rho$ | $m$ | $k$ | epochs | Test Error |
|---|---|---|---|---|---|---|
| WRN40x10 CIFAR100 | AutoDrop | **0.95** | **10** | **64** | **200** | **19.41 ± 0.10** |
| | | – | 5 | – | 200 | 19.84 ± 0.21 |
| | | – | 20 | – | 200 | 23.59 ± 0.22 |
| | | – | 30 | – | 200 | 25.65 ± 0.17 |
| | | – | 50 | – | 200 | 28.72± 0.42 |

Table 13: Ablation study on parameter $m$ for AutoDrop on task WRN40x10/CIFAR100.

| Model | Method | $\rho$ | $m$ | $k$ | epochs | Test Error |
|---|---|---|---|---|---|---|
| WRN40x10 CIFAR100 | AutoDrop | **0.95** | **10** | **64** | **200** | **19.41 ± 0.10** |
| | | 0.5 | – | – | 200 | 25.58±0.46 |
| | | 0.8 | – | – | 200 | 21.03±0.54 |
| | | 0.9 | – | – | 200 | 19.96±0.12 |
| | | 0.99 | – | – | 200 | 30.23±0.35 |

Table 14: Ablation study on parameter $\rho$ for AutoDrop on task WRN40x10/CIFAR100.

## 10.3 ABLATION STUDY FOR $k$

Regarding the sliding window size $k$ used for computing the batch angular velocity, it varies with respect to the size of the training data $N$. Since $k$ decides the frequency of computing the batch angular velocity and we drop the learning rate every time the angular velocity saturates, the learning rate $\alpha_t$ at iteration $t$ for AutoDrop could be simplistically expressed as $\alpha_t = \alpha_0 \rho^{\mathcal{O}(N/k)}$, assuming $\rho$ and $m$ are fixed. Therefore, when the size of the data set $N$ is large, e.g., ImageNet data set

has 14 million images, the sliding window $k$ should be larger than for smaller data sets, such as CIFAR10 and CIFAR100 tasks that have $\sim$10K data points. We found that $k = 64$ performs well for CIFAR10 and CIFAR100 tasks, while $k = 640$ performs much better for ImageNet.

| Model | Ablation Study for $k$ | | | |
|---|---|---|---|---|
| | $k$=32 | $k$=64 | $k$=128 | $k$=256 |
| ResNet18 CIFAR10 | $5.65_{\pm.15}$ | $\mathbf{4.79_{\pm.99}}$ | $6.08_{\pm.11}$ | $7.41_{\pm.24}$ |
| WRN28x10 CIFAR10 | $4.30_{\pm.13}$ | $\mathbf{3.73_{\pm.07}}$ | $5.77_{\pm.13}$ | $7.36_{\pm.15}$ |
| ResNet34 CIFAR100 | $24.07_{\pm.44}$ | $\mathbf{21.82_{\pm.14}}$ | $23.11_{\pm1.3}$ | $28.33_{\pm.20}$ |
| WRN40x10 CIFAR100 | $20.39_{\pm.08}$ | $\mathbf{19.41_{\pm.10}}$ | $24.49_{\pm.16}$ | $28.79_{\pm.32}$ |
| Model | $k$=64 | $k$=256 | $k$=512 | $k$=640 |
| ResNet18 ImageNet | 39.22 | 31.04 | 29.70 | **29.24** |

Table 15: Ablation study for $k$ among different models (test error).

## 11  AUTODROP (APPROXIMATE)

In this section, we analyze why algorithm 2 is an appropriate approximation for Algorithm 2. Note that the main idea behind our algorithm (either Algorithm 1 or 2) is to decrease the learning rate when the angular velocity saturates. Therefore, the key point is how to detect the "saturation". In AutoDrop (Algorithm 1), we determine the saturation of the angular velocity by looking at the difference of the angular velocity in two consecutive epochs. If this difference is smaller than a given threshold $\theta$ then we assume we entered saturation and we will drop the learning rate. However, when it comes to theoretical analysis, it is hard to mathematically measure the "difference" of angular velocities in two consecutive steps and thus the analysis requires some approximations when it comes to defining saturation. Intuitively, when the derivative of the angular velocity is close to zero, we would expect the angular velocity to saturate. This motivates Algorithm 2, which is an approximation to Algorithm 1. Moreover, for the purpose of theoretical analysis, we assume that the angular velocity curve is smooth and could be represented with Equation 12. Under this assumption, the angular velocity is concave with no noise. The behavior of the angular velocity and the learning rate for Algorithm 2 is depicted in Figure 5.

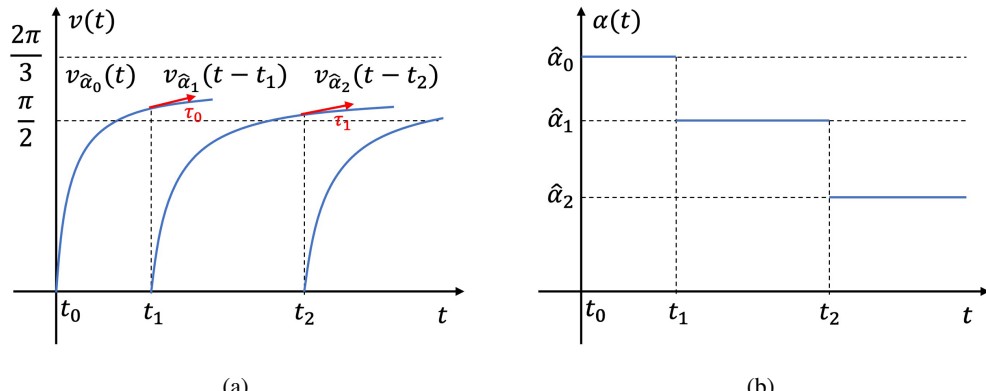

| (a) | (b) |
|---|---|

Figure 5: (a) The behavior of the angular velocity for Algorithm 2. (b) The behavior of the learning rate for Algorithm 2.

## 12  PROOF FOR THEOREM 5.1

Proof in this section in inspired by Yang et al. [2016].

*Proof for Theorem 2.* We denote $\mathcal{G}(x_t; \xi_t) = \mathcal{G}(x_t) = \mathcal{G}_t$. The update formula (9) implies the following recursions:

$$x_{t+1} + p_{t+1} = x_t + p_t - \frac{\alpha_t}{1 - \beta} \mathcal{G}(x_t) \tag{24}$$

$$v_{t+1} = \beta v_t + ((1 - \beta)s - 1)\alpha_t \mathcal{G}(x_t), \tag{25}$$

where $v_t = \frac{1-\beta}{\beta} p_t$ and $p_t$ is given by

$$p_t = \begin{cases} \frac{\beta}{1 - \beta}(x_t - x_{t-1} + s\alpha_{t-1}\mathcal{G}(x_{t-1})), & k \geq 1 \\ 0, & k = 0 \end{cases}. \tag{26}$$

Define $\delta_t = \mathcal{G}_t - \partial f(x_t)$ and let $x^*$ be the optimal point. From the above recursions we have

$$\|x_{t+1} + p_{t+1} - x^*\|^2$$

$$= \|x_t + p_t - x^*\|^2 - \frac{2\alpha_t}{1 - \beta}(x_t + p_t - x^*)^T \mathcal{G}_t + \left(\frac{\alpha_t}{1 - \beta}\right)^2 \|\mathcal{G}_t\|^2$$

$$= \|x_t + p_t - x^*\|^2 - \frac{2\alpha_t}{1 - \beta}(x_t - x^*)^T \mathcal{G}_t - \frac{2\alpha_t \beta}{(1 - \beta)^2}(x_t - x_{t-1})^T \mathcal{G}_t$$

$$- \frac{2s\alpha_t\alpha_{t-1}\beta}{(1 - \beta)^2} \mathcal{G}_{t-1}^T \mathcal{G}_t + \left(\frac{\alpha_t}{1 - \beta}\right)^2 \|\mathcal{G}_t\|^2$$

$$= \|x_t + p_t - x^*\|^2 - \frac{2\alpha_t}{1 - \beta}(x_t - x^*)^T(\delta_t + \partial f(x_t)) - \frac{2\alpha_t \beta}{(1 - \beta)^2}(x_t - x_{t-1})^T(\delta_t + \partial f(x_t))$$

$$- \frac{2s\alpha_t\alpha_{t-1}\beta}{(1 - \beta)^2}(\delta_{t-1} + \partial f(x_{t-1}))^T(\delta_t + \partial f(x_t)) + \left(\frac{\alpha_t}{1 - \beta}\right)^2 \|\delta_t + \partial f(x_t)\|^2. \tag{27}$$

Note that

$$\mathbb{E}[(x_t - x^*)^T(\delta_t + \partial f(x_t))] = \mathbb{E}[(x_t - x^*)^T \partial f(x_t)]$$

$$\mathbb{E}[(x_t - x_{t-1})^T(\delta_t + \partial f(x_t))] = \mathbb{E}[(x_t - x_{t-1})^T \partial f(x_t)]$$

$$\mathbb{E}[(\delta_{t-1} + \partial f(x_{t-1}))^T(\delta_t + \partial f(x_t))] = \mathbb{E}[(\delta_{t-1} + \partial f(x_{t-1}))^T \partial f(x_t)] = \mathbb{E}[\mathcal{G}_{t-1}^T \partial f(x_t)]$$

$$\mathbb{E}[\|\delta_t + \partial f(x_t)\|^2] = \mathbb{E}[\|\delta_t\|^2] + \mathbb{E}[\|\partial f(x_t)\|^2].$$

Taking the expectation on both sides gives the following

$$\mathbb{E}[\|x_{t+1} + p_{t+1} - x^*\|^2]$$

$$= \mathbb{E}[\|x_t + p_t - x^*\|^2] - \frac{2\alpha_t}{1 - \beta}\mathbb{E}[(x_t - x^*)^T \partial f(x_t)] - \frac{2\alpha_t \beta}{(1 - \beta)^2}\mathbb{E}[(x_t - x_{t-1})^T \partial f(x_t)]$$

$$- \frac{2s\alpha_t\alpha_{t-1}\beta}{(1 - \beta)^2}\mathbb{E}[\mathcal{G}_{t-1}^T \partial f(x_t)] + \left(\frac{\alpha_t}{1 - \beta}\right)^2 (\mathbb{E}[\|\delta_t\|^2] + \mathbb{E}[\|\partial f(x_t)\|^2]). \tag{28}$$

Moreover, since f is convex, $\mathbb{E}[\|\mathcal{G}(x; \xi) - \mathbb{E}[\mathcal{G}(x; \xi)]\|] \leq \delta^2$, and $\|\nabla f(x)\| \leq G$, then for any $x$

$$f(x_t) - f(x^*) \leq (x_t - x^*)^T \partial f(x_t)$$

$$f(x_t) - f(x_{t-1}) \leq (x_t - x_{t-1})^T \partial f(x_t)$$

$$- \mathbb{E}[\mathcal{G}_{t-1}^T \partial f(x_t)] \leq \frac{\mathbb{E}[\|\mathcal{G}_{t-1}\|^2 + \|\partial f(x_t)\|^2]}{2} \leq \delta^2/2 + G^2 \leq \delta^2 + G^2$$

$$\mathbb{E}[\|\delta_t\|^2] \leq \delta^2, \quad \mathbb{E}[\|\partial f(x_t)\|^2] \leq G^2.$$

Therefore, (28) can be rewritten as

$$\mathbb{E}[\|x_{t+1} + p_{t+1} - x^*\|^2] \leq \mathbb{E}[\|x_t + p_t - x^*\|^2] - \frac{2\alpha_t}{1 - \beta}\mathbb{E}[f(x_t) - f(x^*)] \tag{29}$$

$$- \frac{2\alpha_t \beta}{(1 - \beta)^2}\mathbb{E}[f(x_t) - f(x_{t-1})] + \frac{2s\beta\alpha_t\alpha_{t-1} + \alpha_t^2}{(1 - \beta)^2}(G^2 + \delta^2).$$

Since $\hat{\alpha}_i$ is decreasing, it implies that $\alpha_t$ is non-increasing. Thus, (30) could be upper-bounded as

$$\mathbb{E}[\|x_{t+1} + p_{t+1} - x^*\|^2] \leq \mathbb{E}[\|x_t + p_t - x^*\|^2] - \frac{2\alpha_t}{1-\beta}\mathbb{E}[f(x_t) - f(x^*)] \tag{30}$$

$$- \frac{2\alpha_t\beta}{(1-\beta)^2}\mathbb{E}[f(x_t) - f(x_{t-1})] + \frac{(2s\beta+1)\alpha_t\alpha_{t-1}}{(1-\beta)^2}(G^2 + \delta^2).$$

Taking $t = 0, ..., T-1$ and $x_{-1} = x_0$, and then summing all the inequalities gives

$$\sum_{t=0}^{T-1}\mathbb{E}[\|x_{t+1}+p_{t+1}-x^*\|^2] \leq \sum_{t=0}^{T-1}\mathbb{E}[\|x_t + p_t - x^*\|^2] - \sum_{t=0}^{T-1}\frac{2\alpha_t}{1-\beta}\mathbb{E}[f(x_t) - f(x^*)]$$

$$- \sum_{t=0}^{T-1}\frac{2\alpha_t\beta}{(1-\beta)^2}\mathbb{E}[f(x_t) - f(x_{t-1})] + \frac{(2s\beta+1)(G^2+\delta^2)}{(1-\beta)^2}\sum_{t=0}^{T-1}\alpha_t\alpha_{t-1}.$$

Therefore,

$$\frac{2}{1-\beta}\sum_{t=0}^{T-1}\alpha_t\mathbb{E}[f(x_t) - f(x^*)] \leq \|x_0 - x^*\|^2 - \|x^T + p_T - x^*\| + \frac{2\beta}{(1-\beta)^2}\sum_{t=0}^{T-1}\alpha_t\mathbb{E}[f(x_{t-1}) - f(x_t)]$$

$$+ \frac{(2s\beta+1)(G^2+\delta^2)}{(1-\beta)^2}\sum_{t=0}^{T-1}\alpha_t\alpha_{t-1},$$

since $\alpha_{T-1} \leq ... \leq \alpha_1 \leq \alpha_0 < 1$, $\min_{t=0,...,T-1}\{\mathbb{E}[f(x_t) - f(x^*)]\} \leq \mathbb{E}[f(x_t) - f(x^*)](\forall t = 0, ..., T-1)$. Then

$$\frac{2}{1-\beta}\min_{t=0,...,T-1}\{\mathbb{E}[f(x_t) - f(x^*)]\}\sum_{t=0}^{T-1}\alpha_t \leq \|x_0 - x^*\|^2 + \frac{2\beta}{(1-\beta)^2}\sum_{t=0}^{T-1}\alpha_t\mathbb{E}[f(x_{t-1}) - f(x_t)]$$

$$+ \frac{(2s\beta+1)(G^2+\delta^2)\sum_{t=0}^{T-1}\alpha_t\alpha_{t-1}}{(1-\beta)^2}.$$

Moreover, $\alpha_t = \hat{\alpha}_i(t_i \leq t < t_{i+1})$ implies that

$$\frac{2}{1-\beta}\min_{t=0,...,T-1}\{\mathbb{E}[f(x_t) - f(x^*)]\}\sum_{t=0}^{T-1}\alpha_t \leq \|x_0 - x^*\|^2 + \frac{2\beta}{(1-\beta)^2}\sum_{i=0}^{n-1}\hat{\alpha}_i\mathbb{E}[f(x_{t_i}) - f(x_{t_{i+1}})]$$

$$+ \frac{(2s\beta+1)(G^2+\delta^2)\sum_{t=0}^{T-1}\alpha_t\alpha_{t-1}}{(1-\beta)^2}.$$

Since $\mathbb{E}[f(x_{t_i}) - f(x_{t_{i+1}})]$ is always upper-bounded by $f(x_0) - f(x^*)$, we have

$$\frac{2}{1-\beta}\min_{t=0,...,T-1}\{\mathbb{E}[f(x_t) - f(x^*)]\}\sum_{t=0}^{T-1}\alpha_t \leq \|x_0 - x^*\|^2 + \frac{2\beta}{(1-\beta)^2}[f(x_0) - f(x^*)]\sum_{i=0}^{n-1}\hat{\alpha}_i$$

$$+ \frac{(2s\beta+1)(G^2+\delta^2)\sum_{t=0}^{T-1}\alpha_t\alpha_{t-1}}{(1-\beta)^2}.$$

After simplification, we have

$$\min_{t=0,...,T-1}\{\mathbb{E}[f(x_t) - f(x^*)]\} \leq \frac{(1-\beta)\|x_0 - x^*\|^2}{2\sum_{t=0}^{T-1}\alpha_t} + \frac{\beta[f(x_0) - f(x^*)]\sum_{i=0}^{n-1}\hat{\alpha}_i}{(1-\beta)\sum_{t=0}^{T-1}\alpha_t}$$

$$+ \frac{(2s\beta+1)(G^2+\delta^2)\sum_{t=0}^{T-1}\alpha_t\alpha_{t-1}}{2(1-\beta)\sum_{t=0}^{T-1}\alpha_t}. \tag{31}$$

Because $\hat{\alpha}_i \le (i+2)^{-1}$, $k_i\hat{\alpha}_i \ge \kappa_1(i+2)^{-\frac{1}{3}}$, $k_i\hat{\alpha}_i\hat{\alpha}_{i-1} \le \kappa_2(i+1)^{-\frac{2}{3}}$, $\forall i = 0, 1, ..., n - 1 (n \gg 1)$,

$$\sum_{i=0}^{n-1} \hat{\alpha}_i \le \sum_{i=0}^{n-1}(i+2)^{-1} = \int_0^{n-1}(i+2)^{-1} = \log(n+1) - \log(2) \tag{32}$$

$$\sum_{t=0}^{T-1} \alpha_t = \sum_{i=0}^{n-1} k_i\hat{\alpha}_i \ge \sum_{i=0}^{n-1} \kappa_1 = \kappa_1 n \tag{33}$$

$$\sum_{t=0}^{T-1} \alpha_t\alpha_{t-1} \le \sum_{i=0}^{n-1} k_i\hat{\alpha}_i\hat{\alpha}_{i-1} \le \kappa_2 \sum_{i=0}^{n-1}(i+1)^{-1} = \kappa_2 \int_0^{n-1}(i+1)^{-1} = \kappa_2 \log n. \tag{34}$$

Substituting (32-34) into inequality (31) gives

$$\min_{t=0,...,T-1}\{\mathbb{E}[f(x_t) - f(x^*)]\} \le \frac{\beta(f(x_0) - f(x^*))[\log(n+1) - \log 2]}{\kappa_1(1-\beta)n} + \frac{(1-\beta)\|x_0 - x^*\|^2}{2\kappa_1 n}$$
$$+ \frac{(2s\beta + 1)(G^2 + \delta^2)\kappa_2 \log n}{2(1-\beta)\kappa_1 n}.$$

$\square$

## 12.1 PROOF FOR THEOREM 5.2

First, we introduce Lemma 12.1 which will be used in the proof for Theorem 5.2. We prove this lemma later in this section.

**Lemma 12.1.** *If sequences $\{\hat{\alpha}_i\}_{i=-1}^{n-1} \subset (0,1)$ and $\{k_i\}_{i=0}^n \subset \mathbb{N}$ satisfy:*

$$\hat{\alpha}_i = (i+2)^{-1}, \quad \frac{\kappa_1}{\hat{\alpha}_i} \le k_i \le \frac{\kappa_2}{\hat{\alpha}_i},$$

*where $\kappa_1$, $\kappa_2$ are constants, then*

$$\hat{\alpha}_i \le (i+2)^{-1}, \quad k_i\hat{\alpha}_i \ge \kappa_1, \quad k_i\hat{\alpha}_i\hat{\alpha}_{i-1} \le \kappa_2(i+1)^{-1}, \quad \forall i = 0, 1, ..., n - 1. \tag{35}$$

*Moreover, suppose $T = \sum_{i=0}^{n-1} k_i$. If $n \gg 1$ the following holds*

$$\frac{\kappa_1 n(n+3)}{2} \le T \le \frac{\kappa_2 n(n+3)}{2}. \tag{36}$$

*Proof for Theorem 5.2.* The derivative of the angular velocity model is:

$$v'_\alpha(t) = \frac{\pi(1 + \epsilon\alpha)}{2\gamma\alpha(t + 1/\gamma\alpha)^2}.$$

Define the gaps of partition $\Pi : 0 = t_0 < t_1 < ... < t_n = T$ derived from the Algorithm 2 as

$$k_i = t_{i+1} - t_i, \quad \forall i = 0, ..., n - 1.$$

Since we drop the learning rate every time the derivative of the angular velocity is smaller that the threshold $\tau_i = \min\{\tau_0, \gamma\hat{\alpha}_i/2\}$, we have

$$v'_{\hat{\alpha}_i}(k_i) = \tau_i \implies k_i = (\gamma\hat{\alpha}_i)^{-\frac{1}{2}}\left[\sqrt{\frac{\pi(1 + \epsilon\hat{\alpha}_i)}{2\tau_i}} - (\gamma\hat{\alpha}_i)^{-\frac{1}{2}}\right].$$

i) From $\tau_i = \min\{\tau_0, \gamma\hat{\alpha}_i/2\}$, we have $\tau_i \le \gamma\hat{\alpha}_i/2$. Therefore,

$$k_i \ge (\gamma\hat{\alpha}_i)^{-\frac{1}{2}}\left[\sqrt{\pi(1 + \epsilon\hat{\alpha}_i)} \times (\gamma\hat{\alpha}_i)^{-\frac{1}{2}} - (\gamma\hat{\alpha}_i)^{-\frac{1}{2}}\right]$$
$$= \frac{1}{\gamma\hat{\alpha}_i}\left[\sqrt{\pi(1 + \epsilon\hat{\alpha}_i)} - 1\right]$$
$$\ge \frac{\sqrt{\pi} - 1}{\gamma\hat{\alpha}_i} \tag{37}$$

ii) From $\tau_i = \min\{\tau_0, \gamma\hat{\alpha}_i/2\}$, we have

$$
\begin{aligned}
k_i &\leq (\gamma\hat{\alpha}_i)^{-\frac{1}{2}} \times \sqrt{\frac{\pi(1 + \epsilon\hat{\alpha}_i)}{2\tau_i}} \\
&= (\gamma\hat{\alpha}_i)^{-\frac{1}{2}} \max\left\{\sqrt{\frac{\pi(1 + \epsilon\hat{\alpha}_i)}{2\tau_0}}, \sqrt{\pi(1 + \epsilon\hat{\alpha}_i)}(\gamma\hat{\alpha}_i)^{-\frac{1}{2}}\right\} \\
&= \max\left\{(\gamma\hat{\alpha}_i)^{-\frac{1}{2}}\sqrt{\frac{\pi(1 + \epsilon\hat{\alpha}_i)}{2\tau_0}}, \frac{1}{\gamma\hat{\alpha}_i}\sqrt{\pi(1 + \epsilon\hat{\alpha}_i)}\right\} \\
&\leq \frac{1}{\gamma\hat{\alpha}_i} \max\left\{\sqrt{\frac{\pi(1 + \epsilon\hat{\alpha}_i)}{2\tau_0}}, \sqrt{\pi(1 + \epsilon\hat{\alpha}_i)}\right\}.
\end{aligned}
$$

Since $\epsilon \in (0, \frac{1}{3\hat{\alpha}_0})$ and $\tau_0 < 2$, we could conclude

$$
k_i \leq \frac{1}{\gamma\hat{\alpha}_i} \max\left\{\sqrt{\frac{2\pi}{3\tau_0}}, \sqrt{\frac{4\pi}{3}}\right\} \leq \frac{1}{\gamma\hat{\alpha}_i}\sqrt{\frac{2\pi}{3\tau_0}}.
$$

Combine i) and ii), we have

$$
\frac{\sqrt{\pi} - 1}{\gamma} \times \frac{1}{\hat{\alpha}_i} \leq k_i \leq \frac{1}{\gamma}\sqrt{\frac{2\pi}{3\tau_0}} \times \frac{1}{\hat{\alpha}_i}. \tag{38}
$$

Define $\kappa_1 = \frac{\sqrt{\pi}-1}{\gamma}$ and $\kappa_2 = \frac{1}{\gamma}\sqrt{\frac{2\pi}{3\tau_0}}$. By Lemma 12.1, we have

$$
\hat{\alpha}_i \leq (i+2)^{-1}, \quad k_i\hat{\alpha}_i \geq \kappa_1, \quad k_i\hat{\alpha}_i\hat{\alpha}_{i-1} \leq \kappa_2(i+1)^{-1}, \quad \forall i = 0, 1, ..., n-1. \tag{39}
$$

Then, by combining (39) with Theorem 5.1 we could conclude that the sequence $\{x_t\}_{t=0}^{T-1}$ generated by the Algorithm 2 satisfies

$$
\begin{aligned}
\min_{t=0,...,T-1}\{\mathbb{E}[f(x_t) - f(x^*)]\} \leq{}& \frac{\beta(f(x_0) - f(x^*))[\log(n+1) - \log 2]}{\kappa_1(1-\beta)n} + \frac{(1-\beta)\|x_0 - x^*\|^2}{2\kappa_1 n} \\
&+ \frac{(2s\beta + 1)(G^2 + \delta^2)\kappa_2 \log n}{2(1-\beta)\kappa_1 n}.
\end{aligned} \tag{40}
$$

By Equation (36) in Lemma 12.1 we have that

$$
\frac{\kappa_1 n(n+3)}{2} \leq T \leq \frac{\kappa_2 n(n+3)}{2}.
$$

Therefore

$$
\sqrt{\frac{2T}{\kappa_2}} - 3 \leq n \leq \sqrt{\frac{2T}{\kappa_1}}. \tag{41}
$$

Combining (41) with (40) gives

$$
\begin{aligned}
\min_{t=0,...,T-1}\{\mathbb{E}[f(x_t) - f(x^*)]\} \leq{}& \frac{\beta(f(x_0) - f(x^*))[\log\left(\sqrt{\frac{2T}{\kappa_1}} + 1\right) - \log 2]}{\kappa_1(1-\beta)\left[\sqrt{\frac{2T}{\kappa_2}} - 3\right]} + \frac{(1-\beta)\|x_0 - x^*\|^2}{2\kappa_1\left[\sqrt{\frac{2T}{\kappa_2}} - 3\right]} \\
&+ \frac{(2s\beta + 1)(G^2 + \delta^2)\kappa_2 \log\left(\sqrt{\frac{2T}{\kappa_1}}\right)}{2(1-\beta)\kappa_1\left[\sqrt{\frac{2T}{\kappa_2}} - 3\right]} \\
={}& O\left(\frac{\log T}{\sqrt{T}}\right)
\end{aligned}
$$

$\square$

## 12.2 PROOF FOR LEMMA 12.1

*Proof for Lemma 12.1.* First, we show bounds from (35) one by one:

i) $\hat{\alpha}_i = (i+2)^{-1} \leq (i+2)^{-1}$.

ii) $k_i \hat{\alpha}_i \geq \kappa_1$.

iii) $k_i \hat{\alpha}_i \hat{\alpha}_{i=1} \leq \kappa_2 \hat{\alpha}_{i-1} = \kappa_2(i+1)^{-1} \leq \kappa_2(i+1)^{-1}$.

Secondly, we compute $T = \sum_{i=0}^{n-1} k_i$ according to the definition of $k_i$. Because $n \gg 1$, the sum of the sequence could be treated as an integral:

$$T = \sum_{i=0}^{n-1} k_i \leq \kappa_2 \sum_{i=0}^{n-1} \frac{1}{\hat{\alpha}_i} = \kappa_2 \sum_{i=0}^{n-1} (i+2) = \frac{\kappa_2 n(n+3)}{2},$$

and

$$T = \sum_{i=0}^{n-1} k_i \geq \kappa_1 \sum_{i=0}^{n-1} \frac{1}{\hat{\alpha}_i} = \kappa_1 \sum_{i=0}^{n-1} (i+2) = \frac{\kappa_1 n(n+3)}{2}.$$

$\square$

# 13 EXPERIMENTAL DETAILS

## 13.1 IMAGE CLASSIFICATION

In addition to the SOTA Baseline referred in the main body of the paper, we also evaluated other competitors, including three manual learning rate schedulers (CLR, OneCycle, ExpLR) and two automatic learning rate schedulers (HD and TLR). For CLR Smith [2017] we test with the *textitOneCycle* learning rate policy and *triangular2* learning policy by adjusting the *stepsize* (the number of iterations in half a cycle) for different models as recommended by the authors. For ExpLR [Li and Arora, 2019], we grid search the decay factor from $\gamma = [0.8, 0.9, 0.95, 0.99, 0.999]$. For HD [Baydin et al., 2018] we grid search the hypergradient learning rate $\beta$ from $[1e-3, 1e-4, 1e-5]$ as suggested in the reference paper. For TLR [Retsinas et al., 2022] we set the gap $p$ for updating the learning rate as $0.33$ epoch and bound $c = 1/4$, as recommended by the authors. For AutoDrop, We set $k = 64$, $\rho = 0.95$, and $m = 10$ as referred in Section 4.1.

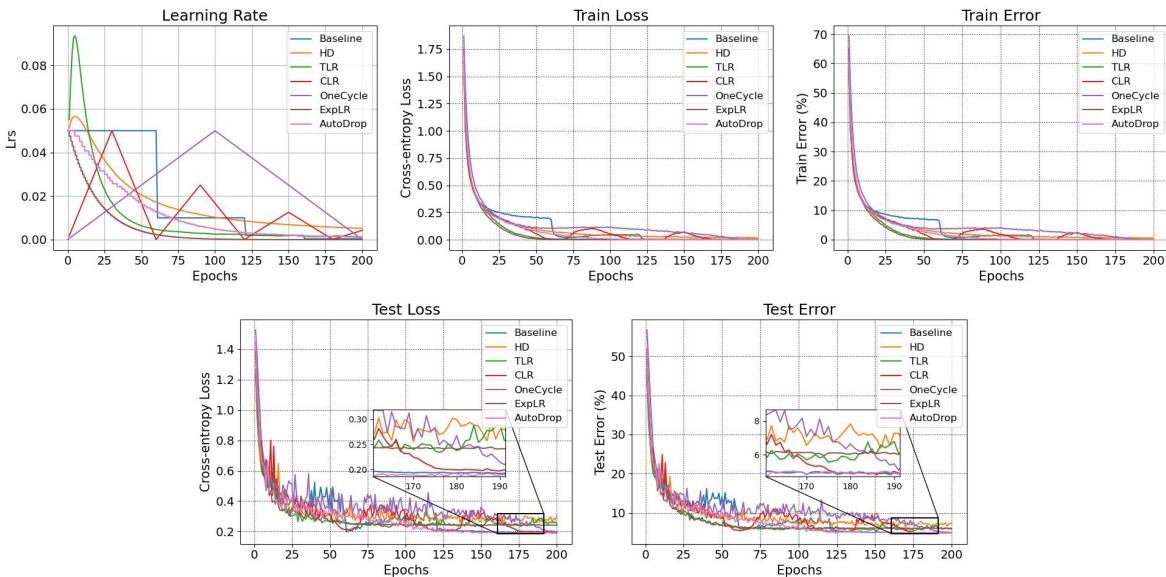

Figure 6: Experimental curves for ResNet18 model and CIFAR-10 data set: learning rate, test loss, test error, and zoomed subplots.

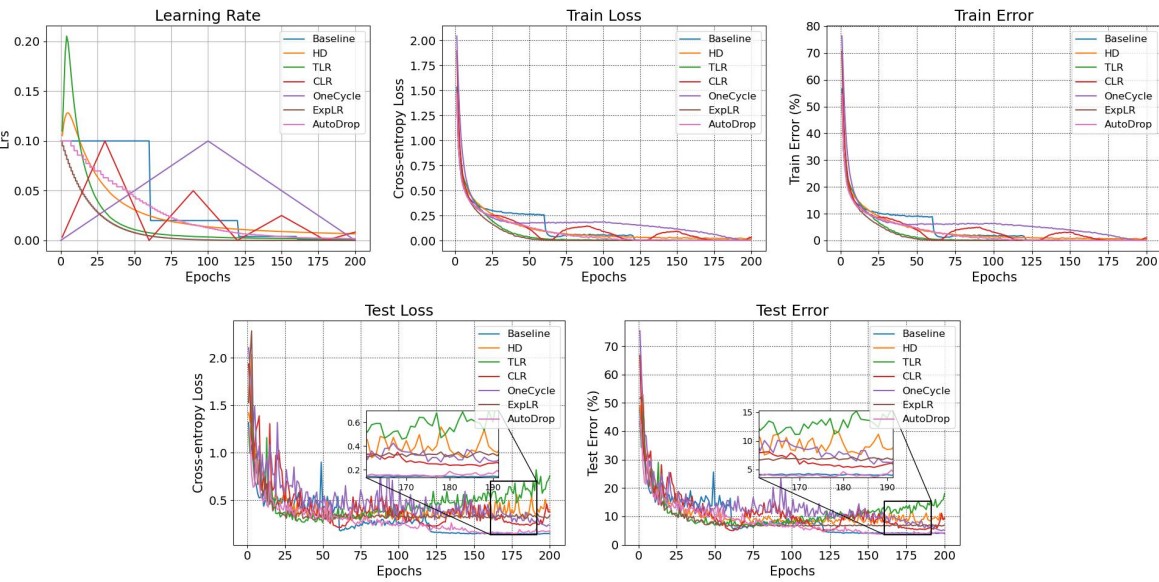

Figure 7: Experimental curves for task WRN28x10/CIFAR-10: learning rate, test loss, test error, and zoomed subplots.

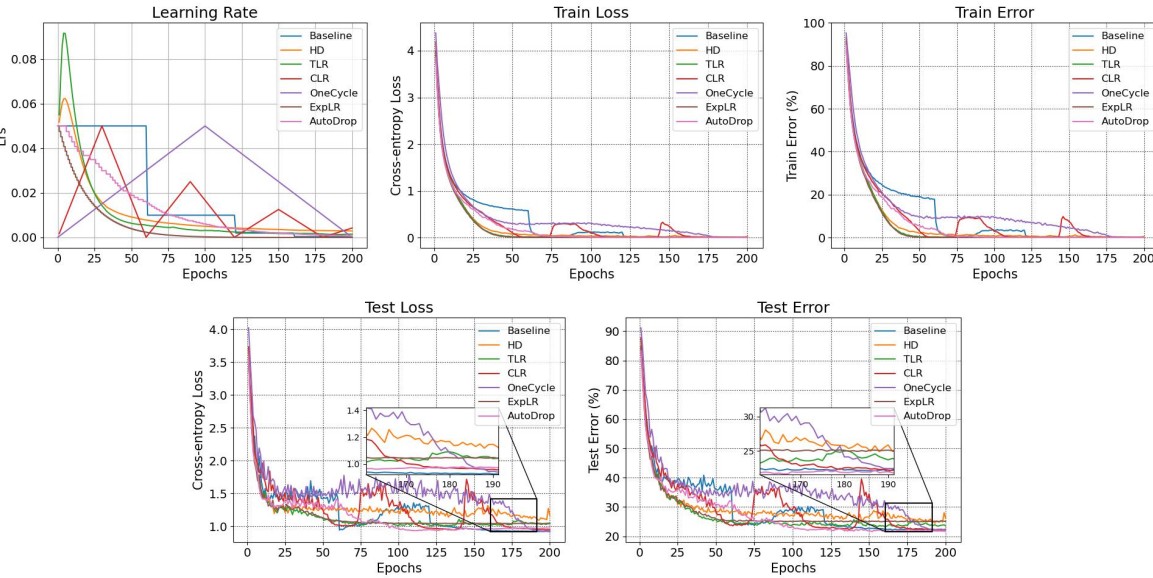

Figure 8: Experimental curves for ResNet34 model and CIFAR-100 data set: learning rate, test loss, test error, and zoomed subplots.

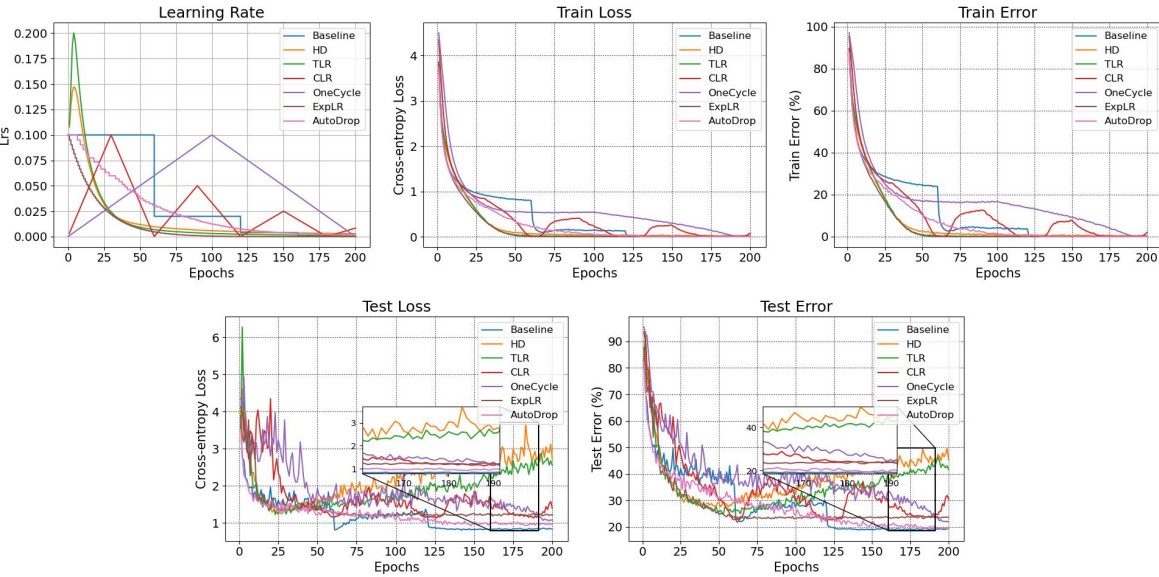

Figure 9: Experimental curves for WRN40x10model and CIFAR-100 data set: learning rate, test loss, test error, and zoomed subplots.

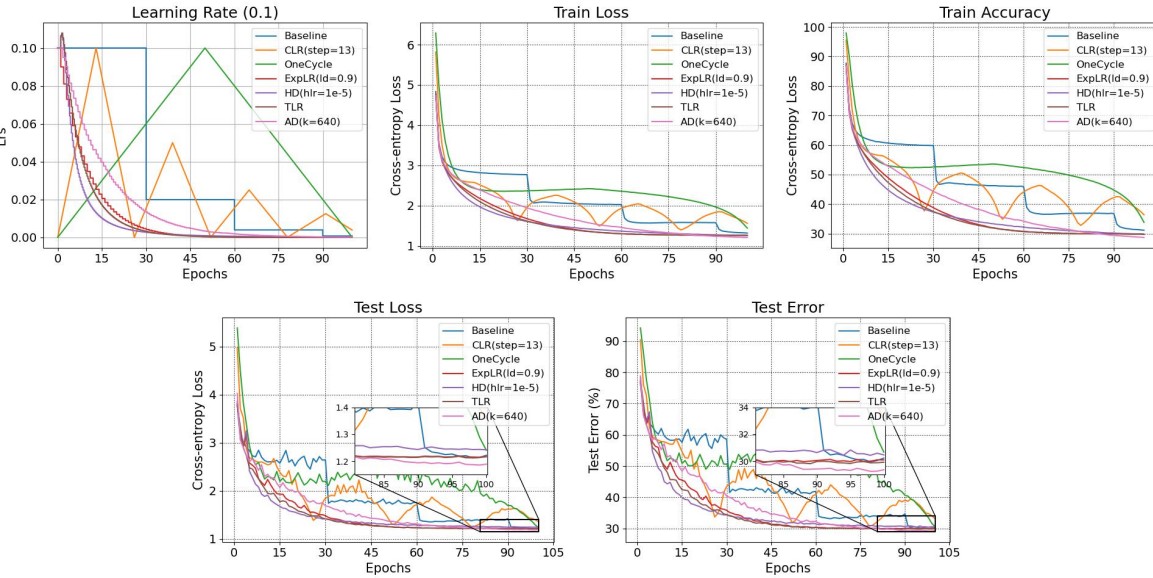

Figure 10: Experimental curves for ResNet18 model and ImageNet data set: learning rate, test loss, test error, and zoomed subplots.

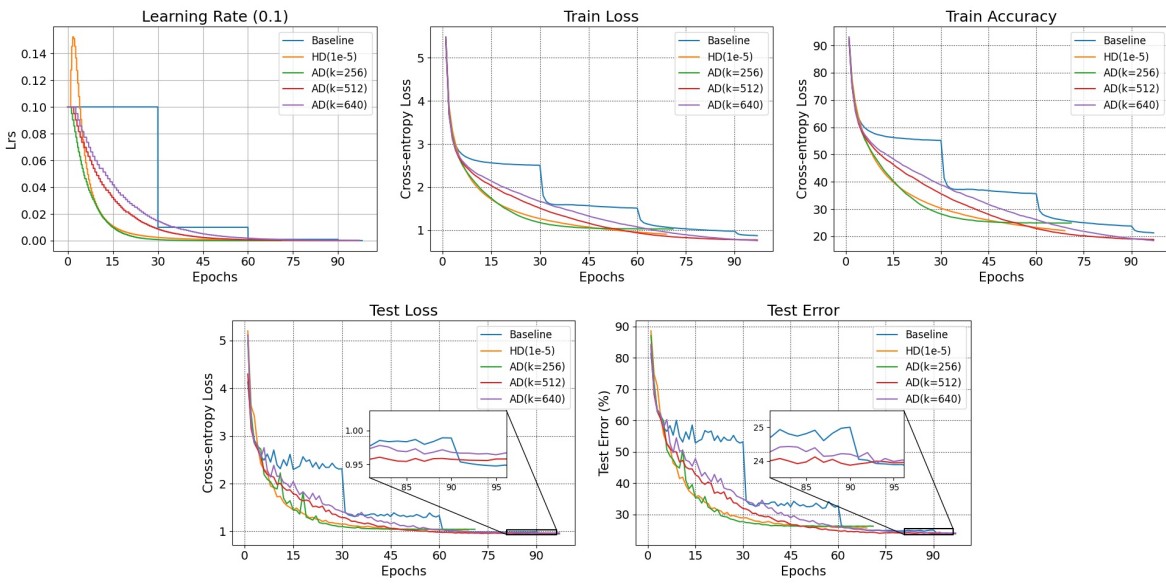

Figure 11: Experimental curves for ResNet50 model and ImageNet data set: learning rate, test loss, test error, and zoomed subplots.

Regarding convergence, note that the theoretical convergence of our method is shown in the paper and the rate in theory matches traditional optimizers, such as SGD. The convergence curves are shown above. The curves reveal that AutoDrop converges to SOTA performance, unlike other methods. Furthermore, the minimum test error for different methods at different epochs (50, 100, 150, 200) for the ResNet18/CIFAR10 task is shown in Table 16. The table demonstrates that AutoDrop reaches comparable performance to SOTA Baseline with sightly faster convergence rate that others cannot attain.

| Test Error | HD | TLR | CLR | OneCycle | ExpLR | SOTA Baseline | AutoDrop |
|---|---|---|---|---|---|---|---|
| 50 epoch | 8.87 | 6.79 | 8.62 | 9.56 | 6.81 | 11.18 | 8.76 |
| 100 epoch | 6.84 | 5.68 | 5.46 | 8.56 | 5.98 | 5.95 | 5.87 |
| 150 epoch | 6.81 | 5.59 | 5.3 | 8.14 | 5.98 | 4.95 | 4.90 |
| 200 epoch | 6.78 | 5.48 | 5.16 | 4.96 | 5.95 | 4.78 | 4.76 |

Table 16: Minimum test error for different methods at different epochs (50, 100, 150, 200) for the ResNet18/CIFAR10.

## 13.2 MACHINE TRANSLATION

A transformer model based on Vaswani et al. [2017] was trained to translate German to English on the WMT2014 data set [Bojar et al., 2014], using ADAM [Kingma and Ba, 2015] optimizer. The performance of our AutoDrop is compared with ReduceLROnPlateau [Red], HD and TLR. We train the model for 10K iterations. Table 4 displays the BLEU score obtained on the test data set. The proposed optimizer led to the highest score on the machine translation task. Figure 12 in Supplementary 13.2 displays the training curve and shows that AutoDrop also converges faster than other methods.

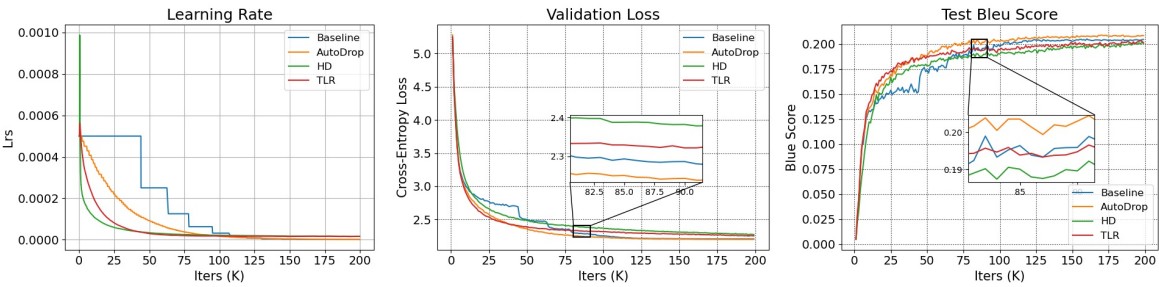

Figure 12: Experimental curves for Transformer and WMT14 data set: learning rate, validation loss, test BLEU score, and zoomed subplots.