# OpenReview forum: "AutoDrop: Training Deep Learning Models with Automatic Learning Rate Drop"
_auai.org/UAI/2024/Conference — UAI 2024 poster_

### Official Review · Reviewer_M4HP · 2024-03-19

**Q2-1 Originality-Novelty:** 3
**Q2-2 Correctness-Technical Quality:** 3
**Q2-5 Clarity Of Writing:** 3

**Q1 Summary And Contributions:**

The authors introduce AutoDrop, an innovative algorithm designed to automatically adjust the learning rate in deep learning (DL) models. AutoDrop leverages the learning dynamics of DL systems, mainly focusing on the pattern of angular velocity in model parameters, which tends to increase rapidly before reaching a soft saturation point where it indicates the optimizer's slowdown. This saturation point serves as the trigger for AutoDrop to reduce the learning rate, allowing the angular velocity to "reset" and repeat the pattern, enhancing training efficiency. Theoretically, the paper contributes by modeling learning rate adjustments based on angular velocity and establishing a general convergence theory for step-wise learning rate adjustments, offering a fresh perspective compared to the traditional continuous analysis.

**Q2-3 Extent To Which Claims Are Supported By Evidence:**

3: Good: the main claims are supported by convincing evidence (in the form of adequate experimental evaluation, proofs, (pseudo-)code, references, assumptions).

**Q2-4 Reproducibility:**

3: Good: key resources (e.g. proofs, code, data) are available and key details (e.g. proofs, experimental setup) are sufficiently well-described for competent researchers to confidently reproduce the main results.

**Q3 Main Strengths:**

Autodrop shows promise as a method for training large neural networks, notably due to its reliance on the angular velocity of the model parameters. Angular velocity serves as a robust measure of an optimizer's convergence dynamics, and its utilization for modulating the learning rate during training can be helpful. The paper is well-structured and provides solid theoretical support for the proposed method.

**Q4 Main Weakness:**

Please check the detailed comments.

**Q5 Detailed Comments To The Authors:**

1.	Definition 1 would be better suited in a different section of the paper to maintain coherence and flow.
2.	The authors' discussion of hyperparameter selection emphasizes computational efficiency; however, their method, while omitting a learning rate schedule, introduces additional hyperparameters. The impact of this on computational cost requires further clarity.
3.	A discrepancy is noted between Section 4.1, which states that variations in model parameters have a mild impact on performance, and Section 10, where certain experiments exhibit up to a 10 percent change in test error. This magnitude of variation could be critical for some problems and warrants further discussion.
4.	The claim in the introduction that autodrop matches or improves the training speed of deep learning models is not supported with empirical evidence. Given that all methods were trained for the same number of epochs, an evaluation of training efficiency is needed.
5.	A comparative analysis of the training times for all methods would be a valuable addition. This comparison would elucidate the computational overhead of each method and is particularly relevant considering the extensive training epochs required by many deep models on large datasets.
6.	An additional recommendation for the authors is to stop training upon the loss's convergence. This strategy would facilitate comparing Autodrop's convergence rate with alternative methods (number of epochs to converge) and permit evaluating the final model on the test set post-convergence.

**Q9 Complying With Reviewing Instructions:**

Yes

---

> ### Author Rebuttal · Authors · 2024-04-08
>
> We thank the Reviewer for his/her feedback. We address specific comments below. We have also already incorporated specific changes into our current draft.
> 1. Note that we introduce the definition of the angular velocity in the very introduction in order to build a clear understanding of what descriptor of the learning dynamics we focus on in the paper. Since our narration early on focuses on angular velocity, we wanted to make sure its definition is put forward right at the very beginning of the paper.
>
> 2. Note that other automatic learning rate schedulers that we compare with (TLR and HD) also have hyper-parameters, as well as all manual learning rate techniques. Let us emphasize that we keep the hyper-parameters fixed across different experiments, as opposed to for example HD method. TLR, similarly to AutoDrop, also does not change the settings of hyper-parameters across different experiments, but their performance is inferior to AutoDrop, and furthermore they perform no ablation studies of their hyper-parameters. We do provide such ablation studies. Regarding computational costs, note that our method does not introduce any additional significant extra computations compared to the existing optimization methods. To support that, see the table below. In this table we report the computational time for a single iteration of HD, TLR, SOTA Baseline, and AutoDrop run on the same machine (NVIDIA GeForce GTX 1080 Ti) for different models on different datasets. We use the same batch size of 64 for all methods to have a fair comparison. As you can see the training time for per-iteration is practically the same for all methods.
> |Training Time/per-iter| HD |TLR |SOTA Baseline|AutoDrop|
> | -------- | ------- | -------- | ------- | -------- |
> | WRN28x10/CIFAR10 |0.21s|0.23s|0.20s|0.20s|
> | WRN40x10/CIFAR100 |0.31s|0.31s|0.29s|0.30s|
> | ResNet50/ImageNet |0.42s|0.43s|0.38s|0.40s|
>
> 3. In Section 10, only extreme cases where $\rho$ or $m$ are set to extremely high values  ($\rho=0.99$, $m=50$) result in significant changes in the error. These settings are however unreasonably high and no practitioner would use those, but we report them for the completeness of the ablation study. We clarified that in our current draft.
>
> 4. We are assuming the Reviewer refers to the sentence from the Introduction stating: “We empirically demonstrate that our method either matches or accelerates the training of DL models and leads to comparable or better generalization compared to SOTA techniques”. We realized that the sentence contained a narrative mistake and instead should be written as “We empirically demonstrate that our method either matches or improves the training of DL models and leads to comparable or better generalization compared to SOTA techniques”. Note that nowhere else in the paper we were claiming computational speed-ups over SOTA, so this was clearly a mistake in writing. Regarding convergence, note that our convergence curves are deferred to the Supplement (Section 13). The curves reveal that AutoDrop converges to SOTA performance, unlike other methods. We emphasize that more in the paper now and make sure we nowhere claim convergence accelerations for AutoDrop.
> Let us please clarify more though. Please, note that our paper does not focus on putting forward an optimizer that accelerates SOTA, but rather focuses on putting forward an automatic learning rate scheduler that can reach SOTA performance without manually tuning the learning rate (note that all SOTA approaches manually tune the learning rate during training). Our method performance-wise matches or outperforms SOTA approach and wins with all other learning rate schedulers, manual and automatic. So for example existing automatic learning rate schedulers, HD and TLR, lose with SOTA since they suffer from the short-horizon problem, which we by design do not have.
>
> 5. We thank the reviewer for the suggestion. Regarding convergence, note that the theoretical convergence of our method is shown in the paper and the rate in theory matches traditional optimizers, such as SGD. The convergence curves are deferred to the Supplement (Section 13). We emphasize that more in the paper now. The curves reveal that AutoDrop converges to SOTA performance, unlike other methods. We also make sure we nowhere claim significant convergence accelerations for AutoDrop in our current draft. Furthermore, see below the minimum test error for different methods at different epochs for the ResNet18/CIFAR10 task. It shows that AutoDrop reaches comparable performance as SOTA Baseline with sightly faster convergence rate that others cannot attain.
> |Test Error| HD |TLR |CLR |OneCycle|ExpLR|SOTA Baseline|AutoDrop|
> | -------- | ------- | -------- | ------- | -------- | ------- | -------- | -------- |
> | 50 epochs |8.87|6.79|8.62|9.56|6.81|11.18|8.76|
> | 100 epochs|6.84|5.68|5.46|8.56|5.98|5.95|5.87|
> | 150 epochs|6.81|5.59|5.3|8.14|5.98|4.95|4.90|
> | 200 epochs|6.78|5.48|5.16|4.96|5.95|4.78|4.76|

---

### Official Review · Reviewer_Z8Kw · 2024-03-22

**Q2-1 Originality-Novelty:** 2
**Q2-2 Correctness-Technical Quality:** 3
**Q2-5 Clarity Of Writing:** 3

**Q1 Summary And Contributions:**

The paper proposes an easy-to-implement automatic learning rate decaying optimizer. The idea is drawn from insights that the optimizer will oscillate around the minima if the training proceeds, and dropping the learning rate at that time may further improve convergence.

**Q2-3 Extent To Which Claims Are Supported By Evidence:**

2: Fair: the main claims are somewhat supported by evidence (but the experimental evaluation may be weak, or does not match entirely with the claims, important baselines may be missing, proofs contain important ideas but lack rigor, algorithmic details are only discussed superficially, references are imprecise, assumptions are not sufficiently motivated or explicated, etc.).

**Q2-4 Reproducibility:**

3: Good: key resources (e.g. proofs, code, data) are available and key details (e.g. proofs, experimental setup) are sufficiently well-described for competent researchers to confidently reproduce the main results.

**Q3 Main Strengths:**

The proposed method is inspired by very elegant insights, which the reviewer greatly appreciates. The method is simple and straightforward to implement, with no significant increase in computational complexity.

**Q4 Main Weakness:**

The theoretical discussion primarily focuses on optimization, while the experimental results report test error rates, which are mainly determined by generalization in overparameterized deep learning settings. Could the authors elaborate on why AutoDrop generalizes better than the state-of-the-art baselines? The reviewer would appreciate either a rigorous theoretical explanation or high-level insights. If the authors intend to highlight the convergence benefits of AutoDrop, a comparison of training speed might be important. Please see the last question below for more details.

**Q5 Detailed Comments To The Authors:**

Questions:
- The reviewer does not fully understand the rationale for dropping the learning rate when the angular velocity saturates. Why does decreasing the learning rate accelerate convergence when the angular velocity saturates?
- The authors claim that AutoDrop decays the learning rate only if the optimizer starts to oscillate around the minima. How does AutoDrop determine if a minima is of good quality? How does it prevent AutoDrop from staying around a suboptimal local minima?
- It is unclear whether AutoDrop would accelerate training in terms of real-world time. Could the authors provide a comparison of wall-clock training time? The reviewer understands the limited time for rebuttal, so it would be sufficient to provide a small set of experiments.

Suggestions:
- If the reviewer understands correctly, the principle behind AutoDrop is to reduce the learning rate when the optimizer's iterate begins to oscillate, with the saturation of angular velocity being the measured indicator. A possible explanation for why AutoDrop works might be that reducing the learning rate alters the stability (or oscillation) conditions of the optimizer, thus bringing the iterate closer to the minimum. Related work such as the concept of the 'edge of stability' [1] could be discussed. Moreover, the notion of bringing the optimizer's iterate closer to the minimum shares similarities with other optimizers like Lookahead [2] and Stochastic Weight Averaging (SWA) [3].

References:

[1] Gradient Descent on Neural Networks Typically Occurs at the Edge of Stability. ICLR, 2022.

[2] Lookahead Optimizer: K Steps Forward, 1 Step Back. NeurIPS, 2019.

[3] Averaging Weights Leads to Wider Optima and Better Generalization. UAI, 2018.

**Q9 Complying With Reviewing Instructions:**

Yes

---

> ### Author Rebuttal · Authors · 2024-04-08
>
> We thank the Reviewer for his/her feedback. We address specific comments below. We have also already incorporated specific changes into our current draft.
> 1. Our paper doesn't aim to propose an optimizer that generalizes better than the SOTA. Instead, it introduces an automatic LR scheduler capable of achieving SOTA performance without manual LR tuning, which is required in all current SOTA approaches. Our method performance-wise matches or outperforms SOTA approach and wins with all other learning rate schedulers, manual and automatic. For example, existing automatic LR schedulers, HD and TLR, lose with SOTA due to the short-horizon problem, which our method avoids by design.
> Our paper demonstrates the theoretical convergence of AutoDrop, the convergence rate matches traditional optimizers like SGD, as detailed in the paper. The convergence curves are deferred to the Supplement (Sec. 13). The curves reveal that AutoDrop converges to SOTA performance, unlike other methods. We make sure we nowhere claim significant convergence accelerations for AutoDrop in our current draft. Furthermore, see below the minimum test error for different methods at different epochs for the ResNet18/CIFAR10 task. It shows that AutoDrop reaches comparable performance as SOTA Baseline with slightly faster convergence rate that others cannot attain.
> |Test Error| HD |TLR |CLR |OneCycle|ExpLR|SOTA Baseline|AutoDrop|
> | -------- | ------- | -------- | ------- | -------- | ------- | -------- | -------- |
> | 50 epochs |8.87|6.79|8.62|9.56|6.81|11.18|8.76|
> | 100 epochs|6.84|5.68|5.46|8.56|5.98|5.95|5.87|
> | 150 epochs|6.81|5.59|5.3|8.14|5.98|4.95|4.90|
> | 200 epochs|6.78|5.48|5.16|4.96|5.95|4.78|4.76|
>
> 2. Note that our method does not introduce any additional significant extra computations compared to the existing methods (see the table below). We report the computational time for a single iteration of HD, TLR, SOTA Baseline, and AutoDrop run on the same machine (NVIDIA GeForce GTX 1080 Ti) for different models on different datasets. We use the same batch size of 64 for all methods to have a fair comparison. As you can see the training time for per-iteration is practically the same for all methods.
> |Training Time/per-iter| HD |TLR |SOTA Baseline|AutoDrop|
> | -------- | ------- | -------- | ------- | -------- |
> | WRN28x10/CIFAR10 |0.21s|0.23s|0.20s|0.20s|
> | WRN40x10/CIFAR100 |0.31s|0.31s|0.29s|0.30s|
> | ResNet50/ImageNet |0.42s|0.43s|0.38s|0.40s|
>
> 3. Dropping the learning rate is not so much to directly accelerate convergence, but rather to help the optimizer that is stuck in the local optimum to escape it, and finally converge to a better quality one. The concept that dropping the learning rate can help convergence is well-described in the literature. Popular manual learning rate methods (linear, stepwise, cosine annealing, exponential, etc.) all decrease the learning rate using different mechanisms. Our mechanism relies on tracking the angular velocity. We found that the saturation of the angular velocity can signal the need to decrease the learning rate, indicating the optimizer's slowdown or entry into a local optimum. Tracking angular velocity saturation is more practical than tracking loss saturation for many reasons (see Fig. 1&3): i) angular velocity curves follow harder saturation pattern, ii) the loss does not necessarily need to have a bounded range, unlike the angular velocity, iii) angular velocity often saturates slightly ahead of the loss function, enabling earlier detection of entering a local optimum. We've further clarified this in the paper by expanding on the argumentation in Sec. 3.
> 4. From the motivation section (Sec. 3), when the algorithm oscillates around the suboptimal local minimum (the training loss no longer decreases), the angular velocity saturates to a stable level. At this point, AutoDrop will decrease the learning rate to escape from the current local minimum. By following the gradient, it converges toward a better minimum with a lower loss value. Results from the experimental section support that employing this strategy enables AutoDrop to eventually converge to a high-quality optimum with SOTA performance.
> 5. The Reviewer's understanding is fundamentally correct, and we appreciate the suggested related works. While the relationship between Hessian information and learning rate introduced in [1] is an interesting topic, it's important to note that the 'edge of stability' discussed applies solely to full-batch gradient descent, and its extension to stochastic optimization remains unclear. As for the Lookhead [2] and SWA [3], our method discuss a different problem. While Lookahead [2] and SWA [3] compute exponential and weighted averages for consecutive gradient steps before updating, our method introduces a novel automatic LR scheduler without altering gradient computations. AutoDrop can therefore be applied on the top of Lookahead and SWA. We are certainly open to consider such extensions in our future works.

---

### Official Review · Reviewer_d2cj · 2024-03-25

**Q2-1 Originality-Novelty:** 2
**Q2-2 Correctness-Technical Quality:** 3
**Q2-5 Clarity Of Writing:** 3

**Q1 Summary And Contributions:**

The paper introduces AutoDrop, a technique for automatically adjusting the learning rate based on angular velocity. Angular velocity represents the angle between successive updates of model parameters. It is inspired by the finding that angular velocity initially rises sharply before reaching a soft saturation. The paper suggests that this saturation in angular velocity serves as a reliable indicator for reducing the learning rate. AutoDrop leverages this concept, lowering the learning rate whenever angular velocity saturates.

**Q2-3 Extent To Which Claims Are Supported By Evidence:**

3: Good: the main claims are supported by convincing evidence (in the form of adequate experimental evaluation, proofs, (pseudo-)code, references, assumptions).

**Q2-4 Reproducibility:**

2: Fair: key resources (e.g. proofs, code, data) are unavailable but key details (e.g. proof sketches, experimental setup) are sufficiently well-described for an expert to confidently reproduce the main results.

**Q3 Main Strengths:**

The primary strengths of the work can be summarized as follows:

1. The utilization of angular momentum as a gauge for determining when to adjust the learning rate presents an intriguing concept, adding novelty to the approach.
2. The method proposed is straightforward yet supported by both theoretical and extensive empirical results.

**Q4 Main Weakness:**

The main weaknesses of the work are outlined as follows:

1. Algorithm 1 introduces numerous hyperparameters. Towards the end of the second paragraph in section 4.1, the author mentions that "the changes of the model performance are mild, i.e., of the order 2.5% − 4%" with hyperparameter tuning, as elaborated in section 10.1. This indicates that careful tuning is necessary for the proposed method to achieve good performance. This raises doubts about whether the proposed method outperforms common approaches like the cosine learning rate scheduler, which may require less tuning effort while achieving comparable performance.

2. Despite the authors' claim in the abstract that the method is computationally cheap, the paper lacks comparisons of the computational cost of the proposed method with other baselines. Table 2 demonstrates that AutoDrop's performance does not exceed that of SOTA Baseline. It is essential to compare computational costs to determine whether AutoDrop truly offers greater efficiency than other methods.

3. In Section 5.2, the model introduced in Equation 12 is stated to adhere to Property P2, including point (iv) which asserts that when the learning rate is sufficiently low, the angular velocity saturates at $π/2$. However, in Figure 3 with the RESNET18 model, this conclusion is contradicted, as the angular velocity may not reach 90 degrees as claimed by the author. This prompts questions about whether the theoretical analysis may not be fulfilled in DL model circumstances.

**Q5 Detailed Comments To The Authors:**

Please see the weaknesses above.

**Q9 Complying With Reviewing Instructions:**

Yes

---

> ### Author Rebuttal · Authors · 2024-04-08
>
> We thank the Reviewer for his/her feedback. We address specific comments below. We have also already incorporated specific changes into our current draft.
> 1. The method is indeed not hyper-parameter free and note that we admit it in the paper and include a separate section discussing the hyper-parameters and performing ablation studies on them (Section 4.1). Other automatic learning rate schedulers that we compare with (TLR and HD) also have hyper-parameters, as well as all manual learning rate techniques. We emphasize however that we use fixed settings of hyper-parameters across different experiments, as opposed to for example HD method. TLR, similarly to AutoDrop, also does not change the settings of hyper-parameters across different experiments, but their performance is inferior to AutoDrop, and furthermore TLR performs no ablation studies of their hyper-parameters. We now removed the phrase "the changes of the model performance are mild, i.e., of the order 2.5% − 4%" in the paper and emphasized more the above.
> In order to further strengthen our experimental section, we add the comparator suggested by the Reviewer. See exemplary result below that includes cosine learning rate scheduler (CosineAnnealing) on the experiments with ResNet. As expected, SOTA Baseline has a better performance than CosineAnnealing. AutoDrop is a winning strategy.
> |Resnet18-CIFAR10 | HD |TLR |CLR |OneCycle|ExpLR|CosAnnealingLR|SOTA Baseline|AutoDrop|
> | -------- | ------- | -------- | ------- | -------- | ------- | -------- | ------- | ------- |
> | Test Error | 6.78 ± 0.23 |5.70 ± 0.19|5.14 ± 0.11|4.86 ± 0.12|5.82 ± 0.10|4.83±0.11|**4.79 ± 0.17** |**4.79 ± 0.99**|
>
> 2. We want to emphasize that our goal is to design the automatic learning rate scheduler that could reach the SOTA performance. We did not intend to outperform the SOTA, but rather show that it is possible to design an automatic learning rate scheduler that indeed can match manual schemes that the SOTA relies on. Our method performance-wise matches or outperforms SOTA approach and wins with all other learning rate schedulers, manual and automatic. So for example existing automatic learning rate schedulers, HD and TLR, lose with SOTA since they suffer from the short-horizon problem, which we by design do not have. Regarding computational costs, note that our method does not introduce any additional significant extra computations compared to the existing optimization methods. To support that, see the table below. In this table we report the computational time for a single iteration of HD, TLR, SOTA Baseline, and AutoDrop run on the same machine (NVIDIA GeForce GTX 1080 Ti) for different models on different datasets. We use the same batch size of 64 for all methods to have a fair comparison. As you can see the training time for per-iteration is practically the same for all methods.
> |Training Time/per-iter| HD |TLR |SOTA Baseline|AutoDrop|
> | -------- | ------- | -------- | ------- | -------- |
> | WRN28x10/CIFAR10 |0.21s|0.23s|0.20s|0.20s|
> | WRN40x10/CIFAR100 |0.31s|0.31s|0.29s|0.30s|
> | ResNet50/ImageNet |0.42s|0.43s|0.38s|0.40s|
> 3. Note that Equation (12) is universal and accommodates any saturation level between 90 and 120 degrees, thus the behavior of the DL model from Figure 3 could very well be represented using this equation. Let us explain better the mechanism that justifies the difference in the behavior between NQM and DL model. The reason DL model does not approach 90 degrees that instead the Noise quadratic Model (NQM) can achieve is that the loss surface for NQM is quadratic convex and DL models instead have a highly non-convex loss surfaces, which makes it terribly difficult to find the global optimum with loss 0. However, note that the saturation levels for the DL model, similarly to NQM, adhere to the range [90, 120] and our Equation (12) can represent any saturation value from this range.

---

### Official Review · Reviewer_KiSd · 2024-03-27

**Q2-1 Originality-Novelty:** 3
**Q2-2 Correctness-Technical Quality:** 3
**Q2-5 Clarity Of Writing:** 3

**Q1 Summary And Contributions:**

This paper proposes a novel algorithm, AutoDrop, to automatically drop the learning rate based on a novel measure of the velocity of the changes of the convergence direction called angular velocity saturation. Both empirical and theoretical properties of angular velocity are studied. The theoretical analysis of the convergence of AutoDrop is also provided. The experiments show that AutoDrop achieves comparable results as the previous learning rate schedulers.

**Q2-3 Extent To Which Claims Are Supported By Evidence:**

2: Fair: the main claims are somewhat supported by evidence (but the experimental evaluation may be weak, or does not match entirely with the claims, important baselines may be missing, proofs contain important ideas but lack rigor, algorithmic details are only discussed superficially, references are imprecise, assumptions are not sufficiently motivated or explicated, etc.).

**Q2-4 Reproducibility:**

3: Good: key resources (e.g. proofs, code, data) are available and key details (e.g. proofs, experimental setup) are sufficiently well-described for competent researchers to confidently reproduce the main results.

**Q3 Main Strengths:**

1. This paper proposes a novel algorithm, AutoDrop, to automatically drop the learning rate based on a novel measure of the velocity of the changes of the convergence direction called angular velocity saturation.
2. Both empirical and theoretical properties of angular velocity are studied.
3. The theoretical analysis of the convergence of AutoDrop is also provided.
4. The experiments show that AutoDrop achieves comparable results as the previous learning rate schedulers.

**Q4 Main Weakness:**

1. The convergence analysis is limited to convex functions.

2. When using AutoDrop, there are even more hyperparameters to be tuned compared to the baselines, which doesn't really seem very automatic to me. Besides, it is unclear some other values in Algorithm 1 are also tunable hyperparameters ("10" in "len(B) >= 10", and "0.1" in "C_i - C_{i-1} < 0.1")

3. Some important baseline is missing, such as cosine learning rate scheduler:
Loshchilov, I., & Hutter, F. SGDR: Stochastic Gradient Descent with Warm Restarts. ICLR 2017

**Q5 Detailed Comments To The Authors:**

1. Is it possible to extend the theoretical analysis to non-convex but smooth functions?

2. In Algorithm 1, are the following values also tunable hyperparameters: "10" in "len(B) >= 10", and "0.1" in "C_i - C_{i-1} < 0.1"? Why making them constant in the algorithm? Is there any theoretical analysis to support making them constant?

3. Cosine learning rate scheduler is widely used and shows good performance in various applications of machine learning. Why not include cosine learning rate in the baselines?

**Q9 Complying With Reviewing Instructions:**

Yes

---

> ### Author Rebuttal · Authors · 2024-04-08
>
> We thank the Reviewer for his/her feedback. We address specific comments below. We have also already incorporated specific changes into our current draft.
> 1. Note that even in the convex case our analysis is highly non-trivial. Allow us to re-emphasize the argument from the paper: All proofs for SGD-based methods require the learning rate to decrease continuously ([1] [2] [3]). On the other hand, we know that SGD does not converge under a constant learning rate. Discrete learning rate policy (as in AutoDrop) covers the space between constant and continuous learning rate decays. It is non-trivial to see that moving away from a continuous learning rate scheme to a step-wise constant scheme will still sustain the rate of convergence the same as in the continuous learning rate techniques. We also show technical conditions capturing the intuitive argument that extremely lazy changes to the learning rate would bring the step-wise constant learning rate scheme close to the constant learning rate method, essentially preventing convergence. We develop a general proof technique (Thm. 5.1) that not only supports AutoDrop, but is also applicable to any learning rate schedulers that decrease the learning rate step-wise. Furthermore, note that the techniques we compare with typically come with no firm theoretical description. We will investigate possible extensions to the non-convex analysis, though note that it is unclear how to prove such convergence guarantee on the top of the discrete learning rate scheduler, and to the best of our knowledge no such proof exists in the literature. For example, using standard analysis of SGD for nonconvex cases, as for example in [2], is not directly possible as bounds in their analysis (e.g. in their Thm. 3) do not hold when the learning rate is changed discretely.
>
>     [1] SGD: General analysis and improved rates
>
>     [2] Unified Convergence Analysis of Stochastic Momentum Methods for Convex and Non-convex Optimization
>
>     [3] ADAM: A METHOD FOR STOCHASTIC OPTIMIZATION
>
> 2. The method isn't hyper-parameter free, as we admitted in the paper, where we include a separate section (Sec. 4.1) discussing and performing ablation studies on hyper-parameters. Phrase “automatic” refers to the techniques that do not need manual adjustments of the learning rate during the optimization process. Other automatic LR schedulers we compare with (TLR and HD) also have hyper-parameters, as do all manual LR techniques. We want to re-emphasize however that in AutoDrop, we keep the hyper-parameters fixed across different experiments, as opposed to for example HD, and report ablation studies justifying the settings of the hyper-parameters that we use. Finally, TLR also does not require hyper-parameters to be changed across different experiments, but their performance is inferior to AutoDrop, and furthermore they perform no ablation studies of their hyper-parameters.
> We then comment on the two conditions mentioned by the Reviewer. The condition “len(B) >10” means that we will not smooth the angular velocity at the very beginning of the training or right after dropping the learning rate - so this is just a common-sense initial condition since we need to gather a few samples before applying smoothing makes sense. As for the condition on "C_i - C_{i-1}," intuitively, the threshold for this term should match the standard deviation of the angular velocity, typically between 0.1 and 0.25 (as shown in the ResNet experiment table below; similar findings were observed in other experiments). We also found that changing this threshold in the range [0.1,0.25] makes no experimental difference. Thus, we set the threshold to 0.1. This is now clarified in the paper.
> |Resnet18-CIFAR10 | 1e-1 |3e-2 |1e-2 |
> | -------- | ------- | -------- | ------- |
> | Standard Deviation  | 0.17 |0.22|0.24|
> 3.  In the selection of SOTA baselines, we choose the best-performing strategy reported in the literature for a given data set and architecture. The best-performing strategy reported by others relies on manual learning rate drop. For Vision tasks the best-performing strategy is referred to as SOTA Baseline and for NLP tasks, this is either ReduceLR or LinearLR in our tables (the references to relevant papers are provided in the paper for each of the tasks). Thus we know we win with the SOTA approaches. In order to further strengthen our experimental section however, we add the comparator suggested by the Reviewer. See exemplary result below that includes cosine learning rate scheduler (CosineAnnealing) on the experiments with ResNet. As expected, SOTA Baseline has a better performance than CosineAnnealing. AutoDrop is a winning strategy.
> |Resnet18-CIFAR10 | HD |TLR |CLR |OneCycle|ExpLR|CosAnnealingLR|SOTA Baseline|AutoDrop|
> | -------- | ------- | -------- | ------- | -------- | ------- | -------- | ------- | ------- |
> | Test Error | 6.78 ± 0.23 |5.70 ± 0.19|5.14 ± 0.11|4.86 ± 0.12|5.82 ± 0.10|4.83±0.11|**4.79 ± 0.17** |**4.79 ± 0.99**|

---

### Meta-Review · Area_Chair_b6V3 · 2024-04-17

The paper proposes an algorithm to automatically decide learning rate drop based on angular velocity of model parameters. It provides theoretical analysis and experimental results to demonstrate the effectiveness of the method.

Strength: The problem is well-motivated and the solution is novel. Both theoretical and empirical results are provided.

Weakness: The analysis only focuses on convex functions. There are many hyper-parameters.

Decision: The paper received consistent reviews. There are concerns about the generality of the theoretical results and algorithm parameters. I encourage the authors to address the concerns in their revision.